# RAG4DMC: Retrieval-Augmented Generation for Data-Level Modality Completion

**Ningxin He**
Nankai University
ningxinhe@mail.nankai.edu.cn

**Yongheng Deng**[*]
Tsinghua University
dyh2024@tsinghua.edu.cn

**Yongjian Fu**
Tsinghua University
yongjianwork@gmail.com

**Sheng Yue**
Sun Yat-sen University
yuesh5@mail.sysu.edu.cn

**Zehui Zhang**
Hangzhou Dianzi University
zhangtianxia918@163.com

**Tiegang Gao**[*]
Nankai University
gaotiegang@nankai.edu.cn

## Abstract

Multi-modal datasets are critical for a wide range of applications, but in practice, they often suffer from missing modalities. This motivates the task of Missing Modality Completion (MMC), which aims to reconstruct missing modalities from the available ones to fully exploit multi-modal data. While pre-trained generative models offer a natural solution, directly applying them to domain-specific MMC is often ineffective, and fine-tuning suffers from limitations like limited complete samples, restricted API access, and high cost. To address these issues, we propose RAG4DMC, a retrieval-augmented generation framework for data-level MMC. RAG4DMC builds a dual knowledge base from complete in-dataset samples and external public datasets, enhanced with feature alignment and clustering-based filtering to mitigate modality and domain shifts. A multi-modal fusion retrieval mechanism combining intra-modal retrieval with cross-modal fusion then provides relevant context to guide generation, followed by a candidate selection mechanism for coherent completion. Extensive experiments on general and domain-specific datasets demonstrate that our method produces more accurate and semantically coherent missing-modality completions, resulting in substantial improvements in downstream image–text retrieval and image captioning tasks.

## 1 Introduction

Multi-modal datasets have become indispensable in advancing a wide range of applications, including vision-language understanding, healthcare analysis, and autonomous systems (Yuan et al., 2025; Xu et al., 2023). However, in practice, such datasets often suffer from missing modalities in some samples, due to reasons such as sensor failure, annotation costs, or data corruption (Jiao et al., 2024). This issue motivates the task of **Missing Modality Completion** (MMC)(Ke et al., 2025), i.e., reconstructing or inferring missing modalities from available ones, thereby enabling more complete and effective utilization of multi-modal data.

With the remarkable success of pre-trained generative models, a natural approach to this task is to leverage them to generate missing modalities conditioned on the observed ones. Nevertheless, directly applying pre-trained generative models to reconstruct domain-specific missing data often yields unsatisfactory results, due to their limited adaptation to specific domains (Ke et al., 2025). A straightforward remedy is to fine-tune these models using the subset of complete samples in the dataset. However, this strategy faces several challenges: (i) the amount of complete data is usually limited, making fine-tuned models prone to overfitting and losing generality; (ii) many pre-trained

---

[*]Corresponding author.

generative models are accessible only through restricted inference APIs, preventing effective fine-tuning; and (iii) fine-tuning large models is often costly and resource-intensive.

To overcome these limitations, we propose a retrieval-augmented generation (RAG) based framework for MMC. RAG combines retrieval from a knowledge base with generative modeling, allowing the model to incorporate external, task-relevant information during generation. While RAG has achieved great success in language modeling and downstream NLP tasks, its potential for MMC remains largely underexplored. In adapting RAG to MMC, two key challenges arise: (i) *how to construct an effective knowledge base from the available multi-modal data*, and (ii) *how to design retrieval strategies that can best augment the generation of missing modalities*.

To this end, we propose RAG4DMC, a retrieval-augmented generation framework tailored for data-level MMC. RAG4DMC leverages a *dual-knowledge-base design* that combines an *internal knowledge base*—built from complete samples in the target dataset—with an *external knowledge base* constructed from publicly available datasets. While the internal knowledge base provides domain-specific information, its limited scale motivates the use of external data as a complementary source, despite domain shifts. To make this integration effective, RAG4DMC encodes all samples into embeddings and introduces *feature alignment mechanisms* to address the embedding misalignment between modalities and knowledge bases. Furthermore, a *clustering-based filtering strategy* is applied to prune irrelevant external samples. On top of the dual knowledge base, RAG4DMC develops a multi-modal fusion retrieval strategy, which combines intra-modal retrieval with cross-modal re-ranking to achieve semantically consistent retrieval results. The retrieved information is used to augment the generative model, which produces multiple candidate completions. Finally, a candidate selection mechanism identifies the most semantically coherent output to fill in the missing modality.

We extensively evaluate RAG4DMC on both general-domain (MSCOCO, Flickr30K) and domain-specific (RSICD) datasets under various modality missing rates. Downstream models trained on datasets completed using RAG4DMC exhibit improvements of up to +5.0 Avg R@1 in image–text retrieval tasks and +5.0 CIDEr in image captioning tasks over those trained on datasets completed via direct generative completion methods.

**The main contributions of this paper are as follows:** (i) To our knowledge, RAG4DMC is the first RAG-based approach to achieve multimodal completion at the raw-data level, generating the actual missing modality instead of relying on latent-feature imputation. (ii) We design an integrated knowledge base combining internal and external samples, equipped with filtering and alignment, and propose a two-stage multi-modal fusion retrieval with candidate selection to mitigate modality gaps and produce semantically faithful generations. (iii) Extensive experiments on MSCOCO, Flickr30K, and RSICD demonstrate that RAG4DMC consistently outperforms baselines, validating both the effectiveness and robustness of the proposed approach.

## 2 RELATED WORK

**Strategies for handling missing modalities.** Existing approaches for handling missing modalities can be broadly categorized into two groups: *non-completion methods* and *completion-based methods*. Non-completion methods avoid explicit imputation and instead rely on fusion strategies or missing-indicator mechanisms to make predictions directly from the observed modalities (Lang et al., 2025; Lee et al., 2023; Guo et al., 2024; Zhao et al., 2024; 2025). While effective in certain scenarios, these methods cannot generate fidelity-guaranteed reusable multimodal samples, which limits their applicability across diverse tasks. In contrast, completion-based methods address the issue by reconstructing the missing modalities before downstream processing. These approaches can be further divided into two subcategories: 1) *Feature-level completion*, which learns to infer missing features in a shared latent space. For example, MMIN (Zhao et al., 2021) imagines latent features and Smil Ma et al. (2021) meta-learns extreme missing patterns. 2) *Data-level completion*, which reconstructs missing modalities directly in the data space. For example, Knowledge Bridger Ke et al. (2025) leverages structured priors from domain knowledge, GTI-MM (Feng et al., 2024) steers diffusion with prompts, and DiCMoR (Wang et al., 2023c) aligns cross-modal distributions via normalising flows. These methods produce more faithful multi-modal datasets that can be fully reused across diverse downstream tasks, but existing methods still suffer from hallucination, poor generalization to rare samples, and heavy resource demands (Wang et al., 2023a). The limitations of existing methods motivate our investigation to RAG-based methods for data-level MMC.

**Multimodal Retrieval-Augmented Generation.** Retrieval-Augmented Generation (RAG) has proven highly effective in NLP for knowledge-grounded text generation and QA (Riedler & Langer, 2024; Lewis et al., 2020; Izacard & Grave, 2020). Recent multimodal extensions adapt RAG to tasks such as captioning, VQA, and reasoning by retrieving relevant images or texts as additional context for generation models (Abootorabi et al., 2025; Lin & Byrne, 2022; Zhang et al., 2024).Despite these advances, the application of RAG to MMC remains underexplored. The most related work is MissRAG(Pipoli et al., 2025), wfhich retrieves prototype representations from the training set to approximate missing inputs. However, MissRAG focuses on feature-level completion at inference, while our approach operates at the data level, explicitly reconstructing missing modalities and generating enriched multi-modal training data.

## 3 METHOD

### 3.1 PROBLEM STATEMENT

We consider the task of data-level MMC in a multi-modal dataset. Let $\mathcal{D} = \{(x_t^I, x_t^T)\}_{t=1}^N$ denote a dataset containing image $(I)$ and text $(T)$ modalities. In practice, many samples are incomplete, i.e., only $x^I$ or $x^T$ is observed. Our goal is to promote a pre-trained generation model $G$ to complete the missing modality $\hat{x}$ from the available one $x$ based on two complementary knowledge bases:

$$\mathcal{K}_{int} = \{(x_i^I, x_i^T)\}_{i=1}^{N_c} \subset \mathcal{D}, \quad \mathcal{K}_{ext} = \{(x_m^I, x_m^T)\}_{m=1}^M,$$

where $\mathcal{K}_{int}$ contains complete samples from the dataset $\mathcal{D}$, and $\mathcal{K}_{ext}$ contains paired image–text samples from publicly available datasets (e.g., CC3M, LAION) to provide additional knowledge. For an incomplete sample $x$, the missing modality is generated as $\hat{x} = G(x, \mathcal{R}(x; \mathcal{K}_{int} \cup \mathcal{K}_{ext}))$, where $\mathcal{R}(\cdot)$ denotes retrieval from the dual knowledge base $\mathcal{K}_{int} \cup \mathcal{K}_{ext}$.

### 3.2 OVERVIEW

The design of RAG4DMC is illustrated in Fig. 1 (algorithm description is illustrated in Appendix A.1). RAG4DMC follows a retrieval-augmented generation pipeline with three key components. It first constructs a dual knowledge base, where the internal base is built from complete samples in the target dataset and the external base is derived from publicly available datasets to compensate for data scarcity. Since embeddings from different modalities and from internal and external sources are often misaligned, RAG4DMC incorporates feature alignment modules that learn mappings to align representations from different modalities and knowledge bases, and applies a clustering-based filtering strategy to remove irrelevant external samples. Given an incomplete sample, the system then performs a two-stage multi-modal fusion retrieval process that first conducts intra-modal retrieval and subsequently refines candidates via cross-modal re-ranking to ensure semantically consistent matches. The retrieved knowledge is used to guide the generative model, which produces multiple candidate completions. Finally, a candidate selection mechanism identifies the most semantically coherent and high-quality output to reconstruct the missing modality.

### 3.3 DUAL KNOWLEDGE BASE CONSTRUCTION

To enable effective retrieval-augmented generation, RAG4DMC constructs a unified dual knowledge base that integrates internal complete samples and external public datasets. Relying solely on the internal dataset is often problematic because the number of complete multi-modal samples is limited and cannot sufficiently cover the diversity of missing cases. For example, in a vision–language dataset collected for autonomous driving, only a fraction of scenes may contain both images and detailed textual annotations, making it difficult to reconstruct missing captions for rare traffic scenarios. By incorporating external public datasets such as CC3M or LAION, the knowledge base can provide broader visual–textual coverage, offering complementary patterns that mitigate the sparsity of internal data. However, directly combining internal and external data introduces three challenges: (i) misalignment between modalities within the same dataset, (ii) noise and irrelevance in external datasets, and (iii) domain shifts across internal and external sources. To address these issues, RAG4DMC employs three key techniques: (i) a cross-modal bidirectional mapping mechanism to reconstruct missing modalities and mitigate modality gaps in the embedding space, (ii) a clustering-based filtering mechanism to prune irrelevant external samples, and (iii) a cross-domain alignment

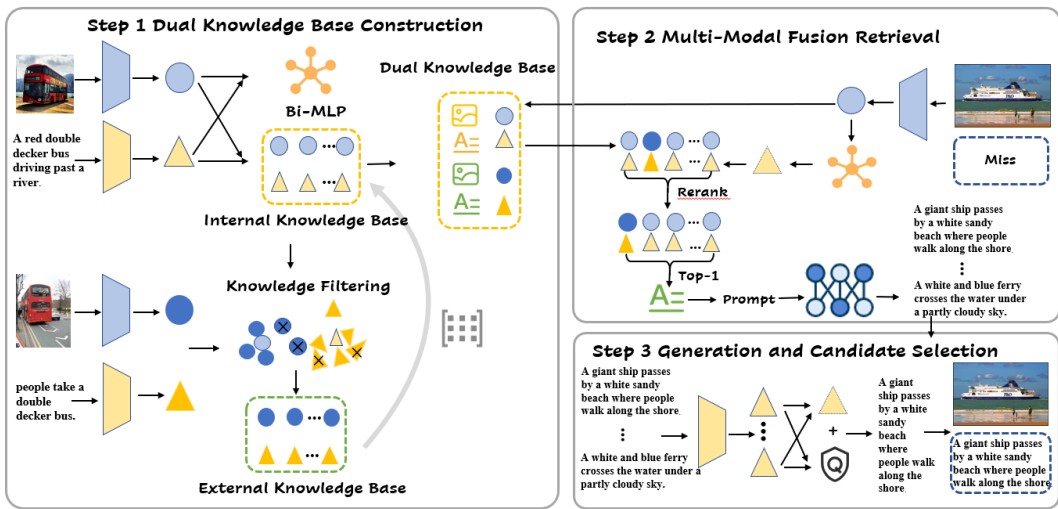

Figure 1: The workflow of RAG4DMC.

strategy to align internal and external embeddings into a unified semantic space. Together, these components yield a semantically consistent multi-modal knowledge base for subsequent retrieval and generation.

**Cross-modal bidirectional mapping.** We use fixed pretrained encoders $E_I : \mathcal{X}^I \to \mathbb{R}^d$ and $E_T : \mathcal{X}^T \to \mathbb{R}^d$ to extract embeddings only for paired multimodal samples. Let $\mathcal{P}_{int} = \{(x_i^I, x_i^T)\}_{i=1}^{N_c} \subset \mathcal{D}$ and $\mathcal{P}_{ext} = \{(x_m^{I,ext}, x_m^{T,ext})\}_{m=1}^{M} \subset \mathcal{D}_{ext}$ denote the sets of complete image–text pairs. For these pairs we compute:

$$z_i^I = E_I(x_i^I), \quad z_i^T = E_T(x_i^T) \quad (i \in \mathcal{P}_{int}) \tag{1}$$

$$z_m^{I,\text{ext}} = E_I(x_m^{I,\text{ext}}), \quad z_m^{T,\text{ext}} = E_T(x_m^{T,\text{ext}}) \quad (m \in \mathcal{P}_{ext}). \tag{2}$$

To bridge the gap between modalities, we train a lightweight MLP for bidirectional cross-modal mapping. We denote the two directions as $f_{I \to T} : \mathbb{R}^d \to \mathbb{R}^d$ and $f_{T \to I} : \mathbb{R}^d \to \mathbb{R}^d$, which share parameters:

$$\mathcal{L}_{\text{map}} = \frac{1}{|\mathcal{P}_{\text{int}}|} \sum_{(x_i^I, x_i^T) \in \mathcal{P}_{\text{int}}} \left( \|f_{I \to T}(z_i^I) - z_i^T\|_2^2 + \|f_{T \to I}(z_i^T) - z_i^I\|_2^2 \right). \tag{3}$$

This bidirectional cross-modal mapping reconstructs missing modalities in the internal embedding space using available ones, and producing pseudo embeddings for retrieval. As the external knowledge base is subsequently aligned into the internal space, training the MLP on internal pairs suffices for both internal and external samples.

**Clustering-based knowledge filtering.** External datasets mix relevant and irrelevant information. To reduce noise, we design a clustering-based knowledge filter. For the image modality, we denote the internal and external image embeddings as $\mathbf{z}^{I,\text{int}}$ and $\mathbf{z}^{I,\text{ext}}$. First, we apply $K$-means to internal and external embeddings to obtain cluster centroids. For $K$ clusters, we compute cluster centroids as the average embedding:

$$\mu_p^{I,\text{int}} = \frac{1}{|c_p^{I,\text{int}}|} \sum_{\mathbf{z}_i^{I,\text{int}} \in c_p^{I,\text{int}}} \mathbf{z}_i^{I,\text{int}}, \quad \mu_q^{I,\text{ext}} = \frac{1}{|c_q^{I,\text{ext}}|} \sum_{\mathbf{z}_m^{I,\text{ext}} \in c_q^{I,\text{ext}}} \mathbf{z}_m^{I,\text{ext}}. \tag{4}$$

where $c_p^{I,\text{int}}$ and $c_q^{I,\text{ext}}$ denote the sets of embeddings assigned to the $p$-th internal cluster and $q$-th external cluster, respectively. Next, we match each external centroid $\mu_q^{I,\text{ext}}$ to its closest internal centroid $\mu_p^{I,\text{int}}$ based on cosine similarity:

$$p_q^I = \arg\max_p \cos\left(\mu_q^{I,\text{ext}}, \mu_p^{I,\text{int}}\right), \tag{5}$$

where $p_q^I$ denotes the index of the internal centroid $\mu_p^{I,\text{int}}$ that is most similar to the external centroid $\mu_q^{I,\text{ext}}$. We then define the corresponding centroid-level and instance-level similarities as:

$$s_{\text{cent}}^I = \cos\left(\mu_q^{I,\text{ext}}, \mu_{p_q^I}^{I,\text{int}}\right), \quad s_{\text{inst}}^I = \cos\left(\mathbf{z}_m^{I,\text{ext}}, \mu_{p_q^I}^{I,\text{int}}\right). \tag{6}$$

where $\mathbf{z}_m^{I,\text{ext}}$ is an external embedding in the $q$-th cluster. Clusters and samples are then filtered by thresholding the similarities:

$$C_{\text{keep}}^I = \{c_q^{I,\text{ext}} \mid s_{\text{cent}}^I \geq \tau_{\text{cent}}\}, \tag{7}$$

$$\mathcal{Z}_{\text{keep}}^I = \{\mathbf{z}_m^{I,\text{ext}} \mid \mathbf{z}_m^{I,\text{ext}} \in c_q^{I,\text{ext}}, \, s_{\text{inst}}^I \geq \tau_{\text{inst}}, \, c_q^{I,\text{ext}} \in C_{\text{keep}}^I\}. \tag{8}$$

We first prune clusters with low centroid similarity and then filter instances within the retained ones to improve efficiency and robustness. The same procedure is applied to text (Appendix A.2). Further analysis of imbalanced clusters and the sensitivity of $\tau_{\text{inst}}$ and $\tau_{\text{cent}}$ is provided in Appendix A.3.

**Cross-domain alignment.** Even after filtering, the internal and retained external embeddings still reside in different semantic spaces due to domain shifts (see t-SNE visualizations in Appendix A.4). Inspired by MUSE (Lample et al., 2018), we adopt an orthogonal Procrustes alignment strategy, performed separately for the image and text modalities using their internal and filtered external embeddings.

For images, let $\{(\mathbf{z}_i^{I,\text{int}}, \mathbf{z}_m^{I,\text{ext}})\}$ denote the mutual nearest neighbor (MNN) pairs identified via CSLS (analysis in Appendix A.6). Stacking these pairs yields matrices $Z_{\text{int}}^I$ and $Z_{\text{ext}}^I$. For texts, the procedure is analogous, producing $Z_{\text{int}}^T$ and $Z_{\text{ext}}^T$. The orthogonal Procrustes problems are then solved:

$$W_I^* = \arg\min_{W \in O(d)} \|Z_{\text{int}}^I - Z_{\text{ext}}^I W\|_F^2, \quad W_T^* = \arg\min_{W \in O(d)} \|Z_{\text{int}}^T - Z_{\text{ext}}^T W\|_F^2, \tag{9}$$

where $O(d) = \{W \in \mathbb{R}^{d \times d} \mid W^\top W = I_d\}$ denotes the set of $d$-dimensional orthogonal matrices. Each admits a closed-form solution via SVD, with derivation in Appendix A.7. The aligned external embeddings are then

$$\tilde{Z}_{\text{ext}}^I = Z_{\text{ext}}^I W_I^*, \quad \tilde{Z}_{\text{ext}}^T = Z_{\text{ext}}^T W_T^*. \tag{10}$$

We iteratively refine $W_I^*$ and $W_T^*$ by alternating between updating MNN pairs using CSLS on the aligned embeddings and recomputing Eq. 9, until convergence, i.e., until $\|W^{(r)} - W^{(r-1)}\|_F < \epsilon$.

After alignment, the internal embeddings remain fixed as the target space, while $W_I^*$ and $W_T^*$ are applied to all external embeddings of the corresponding modality. Each entry in the knowledge base thus contains raw data and its aligned multi-modal embeddings:

$$\mathcal{K}_{\text{int}} = \{x^I, z^I, x^T, z^T\}, \quad \mathcal{K}_{\text{ext}} = \{x^{I,\text{ext}}, \tilde{z}^{I,\text{ext}}, x^{T,\text{ext}}, \tilde{z}^{T,\text{ext}}\}, \tag{11}$$

$$\mathcal{K} = \mathcal{K}_{\text{int}} \cup \mathcal{K}_{\text{ext}}. \tag{12}$$

Thus, modality-level mapping within $\mathcal{D}$ and domain-level alignment across datasets together yield a semantically aligned multimodal knowledge base for retrieval-based completion.

## 3.4 MULTI-MODAL FUSION RETRIEVAL

While RAG has shown strong performance in knowledge-intensive tasks, directly extending it to multimodal missing modality completion poses unique challenges. Conventional approaches often perform cross-modal retrieval (e.g., retrieving images from text), but the inherent modality gap severely limits discriminability, even with powerful joint encoders like CLIP. As shown in Appendix A.4, embeddings of paired image–text samples still occupy distinct subspaces, leading to mismatched candidates and degraded generation quality. This motivates us to design a retrieval mechanism that can both preserve the precision of intra-modal similarity and incorporate cross-modal cues for alignment. To this end, we propose two-stage multi-modal fusion retrieval strategy: (i) intra-modal top-$k$ retrieval for precise candidate selection, and (ii) multimodal re-ranking with pseudo-embeddings of the missing modality. This fusion mitigates the modality gap and yields more semantically aligned candidates for generation. We illustrate the procedure using the case where an image lacks a corresponding text. Let $x_j^I$ be an unpaired image and $z_j^I = E_I(x_j^I)$. We obtain a pseudo-text embedding via the learned map:

$$\hat{z}_j^T = f_{I \to T}(z_j^I). \tag{13}$$

We first conduct intra-modal retrieval. For each candidate $r$ in the knowledge base $\mathcal{K}$ with precomputed embeddings $(z_r^I, z_r^T)$, define

$$s_{\text{img}}(r) = \cos\left(z_j^I, \, z_r^I\right). \tag{14}$$

Let a stable permutation be

$$\sigma = \underset{r}{\text{argsort}} \big( -s_{\text{img}}(r), \, r \big).$$ (15)

The top-$k$ candidate index set is

$$\mathcal{A} = \{\sigma(1), \ldots, \sigma(k)\}, |\mathcal{A}| = k.$$ (16)

For each $r \in \mathcal{A}$, compute the fused score

$$\text{sim}_{\text{fuse}}(r) = \alpha \, \cos(z_j^I, z_r^I) \, + \, (1 - \alpha) \cos(\hat{z}_j^T, z_r^T).$$ (17)

Re-rank stably by the fused score:

$$\pi = \underset{r \in \mathcal{A}}{\text{argsort}} \big( -\text{sim}_{\text{fuse}}(r), \, r \big), r^\star = \pi(1),$$ (18)

and select the top-1 caption $x_{r^\star}^T$. Construct the prompt $\mathcal{P}$ as *"Please write two captions of the image. Caption 1: "$x_{r^\star}^T$". Caption 2:"* and generate sample with the original image $x_j^I$:

$$\tilde{x}^T = G_{I2T}(x_j^I, \mathcal{P}).$$ (19)

The text-only case is symmetric,with detailed formulas provided in Appendix A.5.

## 3.5 GENERATION AND CANDIDATE SELECTION

Given the retrieved exemplars, RAG4DMC leverages modality-specific generators (e.g., $G_{I2T}$ for image-to-text and $G_{T2I}$ for text-to-image) to produce multiple candidate completions. This multi-sample generation increases diversity and reduces the risk of degeneration, but also raises the challenge of identifying the most faithful and coherent output. To address this, we introduce a candidate selection mechanism that jointly evaluates semantic consistency and perceptual quality. For image-only inputs, the caption generator $G_{I2T}$ produces a set of candidate captions $\{\tilde{x}_1^T, \ldots, \tilde{x}_n^T\}$. We score each candidate with a weighted function that balances semantic alignment and linguistic quality:

$$s_T(\tilde{x}^T) = \lambda_1 \cdot \cos \big( E_T(\tilde{x}^T), \hat{z}^T \big) + \lambda_2 \cdot \text{BLEU}(\tilde{x}^T),$$ (20)

where $\hat{z}^T$ is the pseudo-text embedding derived from the input image and BLEU measures n-gram overlap with the retrieved exemplar caption $x_{r^\star}^T$. For text-only inputs, the image generator $G_{T2I}$ produces a set of candidate images $\{\tilde{x}_1^I, \ldots, \tilde{x}_n^I\}$. Each candidate is evaluated by combining semantic similarity and perceptual quality:

$$s_I(\tilde{x}^I) = \lambda_1 \cdot \cos \big( E_I(\tilde{x}^I), \hat{z}^I \big) - \lambda_2 \cdot \text{NIQE}(\tilde{x}^I),$$ (21)

where $\hat{z}^I$ is the pseudo-image embedding from the input text, and NIQE is a no-reference image quality metric (lower is better). The candidate with the highest score is selected as the final completion for the missing modality.

## 4 EVALUATIONS

### 4.1 EXPERIMENTAL SETUP

**Datasets.** We evaluate RAG4DMC on both general-domain and domain-specific datasets. For the general domain, we adopt MSCOCO (Lin et al., 2014) and Flickr30K (Young et al., 2014). For the domain-specific setting, we use the RSICD dataset (Lu et al., 2017). From these datasets, we construct modality-missing dataset. In our experiments, incomplete samples are introduced in the training phase by applying predefined missing rates to each modality. The evaluation phase is conducted exclusively on complete samples. To support retrieval, we adopt external public datasets as knowledge bases. Specifically, CC3M (Sharma et al., 2018) serves as the external knowledge base for MSCOCO and Flickr30K, while NWPU-Caption (Cheng et al., 2022) is used for RSICD. We use BLIP2 (Li et al., 2023) as the text generator and Stable Diffusion XL 1.0 (Rombach et al., 2022) as the image generator across all datasets. We provide more detailed dataset splits and implementation details in the AppendixA.8.

**Metrics.** Following prior work (Ke et al., 2025; Jin et al., 2024), we evaluate the quality of the imputed datasets through their utility in downstream tasks. The underlying rationale is that if the

Table 1: Image–Text Retrieval and Image Captioning results on MSCOCO.

| Method | Image–Text Retrieval | | | | Image Captioning | |
|---|---|---|---|---|---|---|
| | I2T R@1 | T2I R@1 | I2T R@5 | T2I R@5 | CIDEr | BERTScore |
| Complete | 48.9 | 49.6 | 82.3 | 81.0 | 127.9 | 92.6 |
| Drop-Incomplete | 35.4 | 35.1 | 69.5 | 69.3 | 109.2 | 92.1 |
| Direct Generation | 41.4 | 43.9 | 75.4 | 76.3 | 112.2 | 92.2 |
| GTI-MM | 41.0 | 41.7 | 74.9 | 74.9 | 111.0 | 92.2 |
| Knowledge Bridger | 42.5 | 44.5 | 76.1 | 77.5 | 112.2 | 92.2 |
| Vanilla-RAG | 44.9 | 44.6 | 77.9 | 78.0 | 113.4 | 92.3 |
| Combined-RAG | 44.8 | 44.6 | 77.6 | 77.7 | 113.0 | 92.2 |
| KFA-RAG | 45.9 | 46.4 | 78.1 | 78.6 | 115.8 | 92.3 |
| RAG4DMC | **46.6** | **47.5** | **79.0** | **79.7** | **117.2** | **92.5** |

Table 2: Image–Text Retrieval and Image Captioning results on Flickr30K.

| Method | Image–Text Retrieval | | | | Image Captioning | |
|---|---|---|---|---|---|---|
| | I2T R@1 | T2I R@1 | I2T R@5 | T2I R@5 | CIDEr | BERTScore |
| Complete | 53.2 | 52.6 | 81.3 | 81.1 | 71.2 | 91.7 |
| Drop-Incomplete | 45.2 | 47.4 | 75.9 | 75.9 | 64.8 | 91.3 |
| Direct Generation | 47.1 | 49.6 | 76.6 | 77.8 | 65.9 | 91.4 |
| GTI-MM | 46.3 | 48.9 | 75.6 | 76.0 | 65.8 | 91.4 |
| Knowledge Bridger | 47.2 | 49.8 | 77.3 | 78.0 | 66.0 | 91.4 |
| Vanilla-RAG | 48.3 | 50.2 | 77.6 | 78.3 | 67.3 | 91.5 |
| Combined-RAG | 48.3 | 50.2 | 77.3 | 78.0 | 66.9 | 91.5 |
| KFA-RAG | 51.0 | 52.7 | 79.2 | 79.1 | 68.2 | 91.6 |
| RAG4DMC | **52.9** | **53.8** | **80.6** | **81.5** | **70.4** | **91.7** |

imputed data preserves semantic fidelity and consistency with the original multimodal samples, models trained on these datasets should achieve better performance on standard benchmarks. We consider two representative tasks: (i) **Image–text retrieval**, where we train a CLIP (Radford et al., 2021) model on the augmented datasets and evaluate retrieval accuracy using Recall@1 and Recall@5, consistent with prior studies (Hao et al., 2023; Ke et al., 2025; Jin et al., 2024). (ii) **Image captioning**, where we fine-tune LLaVA (Liu et al., 2023) on the augmented datasets to generate textual descriptions, and measure caption quality using CIDEr (Vedantam et al., 2015) and BERTScore (Zhang et al., 2019), following prior work (Li et al., 2022; Wang et al., 2023b; Zhang et al., 2019). For training, CLIP is optimized with AdamW (learning rate $1e-4$, batch size 16) for 20 epochs. LLaVA is fine-tuned with LoRA adapters using AdamW (learning rate $2e-5$, effective batch size 2) for 2 epochs. All evaluations are conducted on the standard test splits: 1,000 samples for MSCOCO and Flickr30K, and 1,093 samples for RSICD.

**Baselines.** Since data-level modality completion based on generative models is largely under-explored, we design the following three categories of baselines: (i) *Naive baselines:* **Drop-Incomplete**, discards all samples with missing modalities. (ii) *Direct generative-based methods:* **Direct Generation**, which fills missing modalities using generative models without retrieval grounding, **GTI-MM**(Feng et al., 2024), which uses generative models to impute missing visual modalities and designs prompts to control the generation quality, and **Knowledge Bridger** (Ke et al., 2025), which extracts knowledge from the available modalities and then guides a generative model to complete the missing ones. (iii) *RAG-based methods:* **Vanilla-RAG**, which builds the knowledge base from internal data only and performs naive cross-modal retrieval; **Combined-RAG**, which builds the knowledge base by directly merging internal and external datasets without filtering or alignment; and **KFA-RAG**, which constructs an integrated knowledge base using the same method in RAG4DMC but performs naive cross-modal retrieval. This design also allows us to dissect the contribution of each component: Combined-RAG vs. Vanilla-RAG highlights the benefit of incorporating external knowledge; KFA-RAG vs. Combined-RAG demonstrates the importance of filtering and alignment in knowledge base construction; and finally, RAG4DMC vs. KFA-RAG validates the effectiveness of our proposed retrieval mechanism.

## 4.2 RESULTS AND ANALYSIS

### 4.2.1 MISSING MODALITY COMPLETION PERFORMANCE

We first evaluate the missing modality completion performance on MSCOCO, Flickr30K, and RSICD datasets. Tables 1–3 summarize performance on downstream image–text retrieval and captioning tasks using datasets completed by different methods. It can be observed that removing incomplete samples (*Drop-Incomplete*) leads to substantial performance degradation. For example,

Table 3: Image–Text Retrieval and Image Captioning results on RSICD.

| Method | Image–Text Retrieval | | | | Image Captioning | |
|---|---|---|---|---|---|---|
| | I2T R@1 | T2I R@1 | I2T R@5 | T2I R@5 | CIDEr | BERTScore |
| Complete | 10.3 | 10.3 | 30.4 | 30.1 | 34.0 | 89.1 |
| Drop-Incomplete | 8.2 | 8.8 | 25.3 | 27.9 | 30.9 | 88.8 |
| Direct Generation | 8.5 | 9.5 | 25.9 | 28.5 | 32.5 | 89.0 |
| GTI-MM | 8.3 | 9.1 | 24.3 | 27.7 | 32.1 | 89.0 |
| Knowledge Bridger | 8.6 | 9.5 | 26.3 | 28.9 | 32.8 | 89.0 |
| Vanilla-RAG | 4.8 | 5.1 | 18.5 | 18.7 | 19.4 | 87.8 |
| Combined-RAG | 4.8 | 5.2 | 18.2 | 18.8 | 18.1 | 87.6 |
| KFA-RAG | 5.3 | 5.6 | 20.2 | 20.3 | 20.6 | 88.1 |
| RAG4DMC | **8.7** | **9.7** | **27.6** | **29.6** | **33.9** | **89.1** |

Avg R@1 on MSCOCO drops from 49.2 to 35.2, which highlights the necessity of modality completion. Introducing retrieval grounding via *Vanilla-RAG* improves performance (+2.2 Avg R@1, +2.1 Avg R@5 on MSCOCO), demonstrating the benefit of exemplar guidance. However, naively adding external data (*Combined-RAG* vs. *Vanilla-RAG*) does not yield gains because of noise and domain mismatch. Filtering and aligning external knowledge (*KFA-RAG* vs. *Combined-RAG*) consistently boosts performance (+1.5 Avg R@1, +0.7 Avg R@5), confirming the effectiveness of our knowledge processing. Finally, integrating multi-modal fusion retrieval in *RAG4DMC* further enhances results (Avg R@1 = 47.1, Avg R@5 = 79.4), approaching the oracle upper bound and validating our retrieval design.

On the domain-specific RSICD dataset, standard RAG variants degrade due to representation bias in the knowledge base. Since CLIP encoders are not trained on remote-sensing data, the extracted image and text features exhibit a modality gap, leading to inaccurate retrieval and reduced completion quality. Comparing *RAG4DMC* with *KFA-RAG* shows that integrating our multi-modal fusion retrieval effectively mitigates this issue and substantially restores performance.

For image captioning, we observe that *Combined-RAG* performs worse than *Vanilla-RAG*, with CIDEr decrease of 1.3% on RSICD dataset. This is because naively merging external samples introduces noise that harms linguistic consistency, captioning requires accurate word-level supervision and is therefore more sensitive to dataset quality. In contrast, *KFA-RAG* alleviates this problem through filtering and alignment, improving CIDEr from 113.0 to 115.8 on MSCOCO. Furthermore, training LLaVA on datasets completed by our method achieves the highest captioning performance, further boosting CIDEr and BERTScore. These results indicate that our completions provide faithful, semantically coherent, and dataset-consistent supervision that benefits not only retrieval but also downstream generative tasks. These findings highlight both the necessity of modality completion and the effectiveness of our proposed design across general and domain-specific datasets.

### 4.3 PERFORMANCE UNDER DIFFERENT MISSING RATES

To evaluate robustness against different degrees of missing information, we conduct experiments on MSCOCO with modality missing rates ranging from 10% to 70%. As shown in Table 4, all methods experience performance degradation as the missing rate increases, but the decline rates differ significantly. Generation-based methods (*Direct Generation*, *GTI-MM*) deteriorate the fastest, reflecting their reliance on generative filling without sufficient grounding, which amplifies errors when larger portions of data are absent. In contrast, *Vanilla-RAG* demonstrates greater stability by leveraging exemplar grounding from internal data, while *KFA-RAG* further enhances robustness by filtering noisy external samples and aligning them with internal exemplars. RAG4DMC consistently surpasses all baselines across missing rates, with particularly pronounced gains under high-missing settings. For example, at 70% missing rate, RAG4DMC improves Avg R@1 by +2.3 over *Vanilla-RAG*, highlighting its ability to deliver faithful and reliable completions even in highly incomplete scenarios. These results demonstrate the effectiveness and robustness of RAG4DMC, showing that its integration of modality completion, retrieval grounding, and knowledge filtering enables consistent performance even under severe data scarcity.

### 4.4 IMPACT OF GENERATION CANDIDATE NUMBER

In RAG4DMC, the generation and candidate selection module introduces a hyperparameter $n$, which determines the number of generated candidates. To investigate its effect, we conduct experiments on MSCOCO with different $n$ values ($n \in \{1, 3, 5, 10\}$). As shown in Table 5, the performance of RAG4DMC consistently improves as $n$ increases. For example, Avg R@1 of RAG4DMC rises from

Table 4: Performance of RAG4DMC on MSCOCO under different modality missing rates.

| MissRate | Method | Image–Text Retrieval | | | | Image Captioning | |
|---|---|---|---|---|---|---|---|
| | | I2T R@1 | T2I R@1 | I2T R@5 | T2I R@5 | CIDEr | BERTScore |
| 0.1 | Drop-Incomplete | 49.3 | 49.1 | 80.7 | 79.7 | 128.2 | 92.4 |
| | Direct Generation | 50.5 | 51.3 | 81.6 | 81.4 | 129.7 | 92.7 |
| | GTI-MM | 49.9 | 49.7 | 80.9 | 80.2 | 128.9 | 92.6 |
| | Knowledge Bridger | 50.9 | 51.5 | 81.9 | 81.9 | 129.5 | 92.6 |
| | Vanilla-RAG | 51.7 | 51.8 | 82.4 | 82.6 | 130.4 | 92.7 |
| | Combined-RAG | 51.6 | 51.8 | 82.3 | 82.4 | 129.7 | 92.6 |
| | KFA-RAG | 52.7 | 52.9 | 83.9 | 83.6 | 131.7 | 92.7 |
| | RAG4DMC | **53.9** | **54.1** | **85.6** | **85.5** | **133.8** | **92.8** |
| 0.3 | Drop-Incomplete | 46.7 | 46.4 | 78.7 | 77.8 | 120.5 | 92.4 |
| | Direct Generation | 47.9 | 48.1 | 80.3 | 80.6 | 121.7 | 92.6 |
| | GTI-MM | 47.5 | 47.8 | 79.9 | 80.0 | 121.1 | 92.5 |
| | Knowledge Bridger | 48.2 | 49.1 | 80.5 | 81.1 | 121.8 | 92.6 |
| | Vanilla-RAG | 48.3 | 49.3 | 81.2 | 81.5 | 123.2 | 92.6 |
| | Combined-RAG | 48.1 | 49.2 | 80.9 | 81.5 | 122.8 | 92.6 |
| | KFA-RAG | 50.2 | 51.7 | 82.3 | 83.0 | 124.7 | 92.7 |
| | RAG4DMC | **52.3** | **52.9** | **84.1** | **84.4** | **125.8** | **92.7** |
| 0.5 | Drop-Incomplete | 42.7 | 45.6 | 76.9 | 78.5 | 116.9 | 92.4 |
| | Direct Generation | 45.5 | 45.8 | 77.2 | 77.9 | 118.5 | 92.4 |
| | GTI-MM | 44.3 | 44.9 | 76.3 | 76.6 | 118.2 | 92.4 |
| | Knowledge Bridger | 45.3 | 45.8 | 77.3 | 78.5 | 118.6 | 92.5 |
| | Vanilla-RAG | 45.8 | 46.1 | 78.3 | 78.9 | 119.8 | 92.5 |
| | Combined-RAG | 45.6 | 46.1 | 78.0 | 78.7 | 119.6 | 92.5 |
| | KFA-RAG | 46.7 | 47.1 | 79.4 | 79.2 | 121.6 | 92.5 |
| | RAG4DMC | **48.4** | **47.7** | **80.1** | **80.2** | **123.6** | **92.6** |
| 0.7 | Drop-Incomplete | 35.4 | 35.1 | 69.5 | 69.3 | 109.1 | 92.1 |
| | Direct Generation | 41.4 | 43.9 | 75.4 | 76.3 | 112.2 | 92.2 |
| | GTI-MM | 41.0 | 41.7 | 74.9 | 74.9 | 111.1 | 92.2 |
| | Knowledge Bridger | 42.5 | 44.5 | 76.1 | 77.5 | 112.2 | 92.2 |
| | Vanilla-RAG | 44.9 | 44.6 | 77.9 | 78.0 | 113.4 | 92.3 |
| | Combined-RAG | 44.8 | 44.6 | 77.6 | 77.7 | 113.0 | 92.2 |
| | KFA-RAG | 45.9 | 46.4 | 78.1 | 78.6 | 115.8 | 92.3 |
| | RAG4DMC | **46.6** | **47.5** | **79.0** | **79.7** | **117.2** | **92.4** |

Table 5: Performance of RAG4DMC on MSCOCO with different generation candidate number $n$.

| $n$ | Image–Text Retrieval | | | | Image Captioning | |
|---|---|---|---|---|---|---|
| | I2T R@1 | T2I R@1 | I2T R@5 | T2I R@5 | CIDEr | BERTScore |
| 1 | 46.6 | 47.5 | 79.0 | 79.7 | 117.2 | 92.4 |
| 3 | 47.1 | 48.0 | 79.4 | 79.8 | 117.9 | 92.4 |
| 5 | 47.4 | 48.2 | 79.8 | 80.0 | 119.3 | 92.5 |
| 10 | **47.7** | **48.5** | **80.1** | **80.3** | **121.5** | **92.6** |

47.1 at $n = 1$ to 48.1 at $n = 10$, while CIDEr rises from 117.2 to 121.5. This demonstrates that RAG4DMC effectively identifies the best candidate among multiple generations, filtering out noisy outputs and ensuring that diversity translates into real performance gains.

## 4.5 Impact of Retrieval Size

We further investigate the effect of the retrieval hyperparameter $k$, which controls the number of exemplars retrieved for RAG. As shown in Table 6, increasing $k$ from 3 to 10 steadily improves performance, since a larger candidate pool increases the likelihood of including highly relevant exemplars, enhancing the quality of generation. However, when $k$ becomes too large (e.g., 15), performance slightly degrades, likely due to the introduction of less relevant or noisy exemplars, which can misguide the generator. This suggests that a moderate number of high-quality retrievals strikes a good balance between diversity and precision. We set $k = 10$ in all main experiments based on this trade-off.

## 4.6 Sensitivity to Knowledge Filtering Thresholds

We conducted additional sensitivity experiments by varying $\tau_{\text{cent}}$ and $\tau_{\text{inst}}$ within a reasonable range (0.6–0.8). The results (Table 7) show that retrieval and captioning performance remains highly stable: the changes in R@1, R@5, CIDEr, and BERTScore are minimal, indicating that our filtering strategy is robust to these thresholds.

Table 6: Performance of RAG4DMC with different retrieval size $k$.

| $k$ | I2T R@1 | T2I R@1 | I2T R@5 | T2I R@5 | CIDEr | BERT Score |
|---|---|---|---|---|---|---|
| 3 | 46.5 | 46.1 | 77.7 | 76.6 | 109.4 | 92.1 |
| 5 | 46.6 | 47.5 | 79.0 | 79.7 | 110.7 | 92.2 |
| 10 | **47.7** | **48.5** | **80.1** | **80.3** | **117.2** | **92.4** |
| 15 | 41.9 | 42.9 | 75.0 | 76.7 | 104.1 | 92.5 |

Table 7: Performance of RAG4DMC with different filtering thresholds.

| Filtering Threshold | I2T R@1 | T2I R@1 | I2T R@5 | T2I R@5 | CIDEr | BERT Score |
|---|---|---|---|---|---|---|
| 0.6 | 46.5 | 47.1 | 79.0 | 79.0 | 117.0 | 92.4 |
| 0.7 | 46.6 | 47.5 | 79.0 | 79.7 | 117.2 | 92.5 |
| 0.8 | **46.9** | **47.6** | **79.6** | **79.9** | **117.2** | **92.5** |

## 4.7 IMPACT OF EXTERNAL KNOWLEDGE BASE SIZE

To investigate how performance varies with the size of the external knowledge base, we conducted experiments on MSCOCO using external samples ranging from 5k to 15k. As shown in Table 8, the results demonstrate a monotonic improvement as the size of the external KB increases, though with diminishing returns at larger sizes. This suggests that while a larger external KB offers richer exemplars and broader coverage, our method does not depend on extremely large corpora to perform effectively—competitive performance is already achieved with just 5k external samples.

Table 8: Performance of RAG4DMC with different number of external knowledge base samples.

| Number of External KB | I2T R@1 | T2I R@1 | I2T R@5 | T2I R@5 | CIDEr | BERTScore |
|---|---|---|---|---|---|---|
| 5000 | 46.0 | 46.6 | 78.6 | 78.8 | 116.4 | 92.4 |
| 10000 | 46.6 | 47.5 | 79.0 | 79.7 | 117.2 | 92.5 |
| 15000 | **47.1** | **48.1** | **79.3** | **80.1** | **117.5** | **92.7** |

## 4.8 COMPLEXITY ANALYSIS

In Appendix A.9., we provide a detailed theoretical complexity analysis for each component in our pipeline. As shown there, the dominant factors are the number of samples $N$, the number of clusters $k$, and the feature dimension $d$. We supplemented experiments measuring the runtime for each module under varying $N$, $k$, and $d$, on an NVIDIA GeForce RTX 3090 GPU.

As summarized in Table 9, under the largest tested configuration ($N = 1k+10k$, $k = 128$, $d = 1024$), the end-to-end pipeline—including knowledge filtering, cross-domain alignment, and bidirectional MLP training—completes in approximately $629\,\mathrm{s}$ (10.48 minutes), while retrieval remains extremely lightweight ($0.0031\,\mathrm{s}$). Thus, although the pipeline involves clustering and nearest-neighbor search, the overall computational overhead is clearly manageable in practice. Table 9 further breaks down the major sources of computation:(i) K-means is most sensitive to $N$ and $k$, consistent with its standard computational behavior; (ii) Cross-domain alignment scales primarily with $d$, since nearest-neighbor search is performed in the feature space; (iii) Bidirectional MLP training dominates the total runtime but is largely insensitive to $N$ and $k$, as it operates only on internal feature representations and is mainly driven by $d$; (iv) Filtering and retrieval incur negligible cost across all configurations. Overall, these results demonstrate that the scaling behaviors of all modules are predictable and well controlled. Importantly, none of the components introduce unexpected or prohibitive overhead. Moreover, users may flexibly adjust $N$, $k$, and $d$ to trade off between efficiency and accuracy, enabling deployment under different computational budgets.

Table 9: Time cost (in seconds) of each module under different settings.

| Configuration | K-means Clustering | Filtering | Cross-Domain Alignment | Cross-Modal Mapping | Multi-Modal Retrieval |
|---|---|---|---|---|---|
| $N = 1k + 5k, k = 128, d = 512$ | 21.43 | 0.30 | 0.40 | 405.17 | 0.0012 |
| $N = 1k + 10k, k = 128, d = 512$ | 39.14 | 0.64 | 0.55 | 407.14 | 0.0014 |
| $N = 1k + 20k, k = 128, d = 512$ | 112.87 | 1.35 | 0.99 | 406.25 | 0.0017 |
| $N = 1k + 10k, k = 64, d = 512$ | 32.98 | 0.58 | 0.52 | 408.95 | 0.0015 |
| $N = 1k + 10k, k = 256, d = 512$ | 46.97 | 0.78 | 0.54 | 408.98 | 0.0015 |
| $N = 1k + 10k, k = 128, d = 768$ | 53.49 | 0.82 | 1.43 | 499.29 | 0.0018 |
| $N = 1k + 10k, k = 128, d = 1024$ | **92.20** | **0.88** | **2.01** | **533.85** | **0.0031** |

## 5 CONCLUSION

This paper proposes RAG4DMC, a retrieval-augmented framework for data-level missing modality completion. Our approach leverages a dual knowledge base and multi-modal fusion retrieval with semantic-quality-based candidate selection to generate faithful and semantically coherent completions. Extensive experiments on both general-domain and domain-specific datasets demonstrate that RAG4DMC consistently outperforms existing baselines, significantly improving the training performance of downstream models on image–text retrieval and captioning tasks.

ACKNOWLEDGMENTS

This research was supported in part by the National Key R&D Program of China under Grant 2024YFB4207203; in part by the National Natural Science Foundation of China under Grant No. 62402267, 62572496, and 52401376; in part by the Shenzhen Science and Technology Program under Grant JCYJ20250604175500001; and in part by the "Pioneer" and "Leading Goose" R&D Program of Zhejiang under Grant 2024C03254.

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

# A APPENDIX

## A.1 THE FRAMEWORK OF RAG4DMC

The complete algorithmic procedures are provided in this section. Specifically, Algorithm 1 describes the construction of the dual knowledge base, while Algorithm 2 outlines the overall inference workflow of multi-modal fusion retrieval and candidate selection.

## A.2 CLUSTERING-BASED KNOWLEDGE FILTERING FOR TEXT EMBEDDINGS

For the text modality, the internal and external text embeddings are denoted as $\mathbf{z}^{T,\text{int}}$ and $\mathbf{z}^{T,\text{ext}}$, and processed analogously to the image modality. We apply $K$-means to obtain cluster centroids, computed as the average embedding:

$$\mu_p^{T,\text{int}} = \frac{1}{|c_p^{T,\text{int}}|} \sum_{\mathbf{z}_i^{T,\text{int}} \in c_p^{T,\text{int}}} \mathbf{z}_i^{T,\text{int}}, \quad \mu_q^{T,\text{ext}} = \frac{1}{|c_q^{T,\text{ext}}|} \sum_{\mathbf{z}_m^{T,\text{ext}} \in c_q^{T,\text{ext}}} \mathbf{z}_m^{T,\text{ext}}. \tag{22}$$

Each external centroid $\mu_q^{T,\text{ext}}$ is matched to its closest internal centroid $\mu_p^{T,\text{int}}$:

$$p_q^T = \arg\max_p \cos\left(\mu_q^{T,\text{ext}}, \mu_p^{T,\text{int}}\right), \tag{23}$$

and we define centroid-level and instance-level similarities as

$$s_{\text{cent}}^T = \cos\left(\mu_q^{T,\text{ext}}, \mu_{p_q^T}^{T,\text{int}}\right), \quad s_{\text{inst}}^T = \cos\left(\mathbf{z}_m^{T,\text{ext}}, \mu_{p_q^T}^{T,\text{int}}\right), \tag{24}$$

where $\mathbf{z}_m^{T,\text{ext}} \in c_q^{T,\text{ext}}$. Clusters and instances are then filtered hierarchically: we first prune clusters with low centroid similarity and then filter embeddings within the retained clusters:

$$C_{\text{keep}}^T = \{c_q^{T,\text{ext}} \mid s_{\text{cent}}^T \geq \tau_{\text{cent}}\}, \tag{25}$$

$$\mathcal{Z}_{\text{keep}}^T = \{\mathbf{z}_m^{T,\text{ext}} \mid \mathbf{z}_m^{T,\text{ext}} \in c_q^{T,\text{ext}}, \ s_{\text{inst}}^T \geq \tau_{\text{inst}}, \ c_q^{T,\text{ext}} \in C_{\text{keep}}^T\}. \tag{26}$$

---

**Algorithm 1** The Framework of Dual Knowledge Base Construction

---

**Require:** Internal data $\mathcal{D}$, external data $\mathcal{D}_{\text{ext}}$; frozen encoders $E_I, E_T$; mapping MLPs $f_{I \to T}, f_{T \to I}$; number of clusters $K$; thresholds $\tau_{\text{cent}}, \tau_{\text{inst}}$; tolerance $\epsilon$
**Ensure:** Multi-modal knowledge base $\mathcal{K}$, trained mapping MLPs $f_{I \to T}, f_{T \to I}$
  1: Construct paired sets: $\mathcal{P}_{\text{int}} = \{(x_i^{I,\text{int}}, x_i^{T,\text{int}})\}$, $\mathcal{P}_{\text{ext}} = \{(x_m^{I,\text{ext}}, x_m^{T,\text{ext}})\}$.
  2: **for** $(x_i^{I,\text{int}}, x_i^{T,\text{int}}) \in \mathcal{P}_{\text{int}}$ **do**
  3:    $z_i^{I,\text{int}} \leftarrow E_I(x_i^{I,\text{int}})$, $z_i^{T,\text{int}} \leftarrow E_T(x_i^{T,\text{int}})$                                  ▷ Eq. 1
  4: **end for**
  5: **for** $(x_m^{I,\text{ext}}, x_m^{T,\text{ext}}) \in \mathcal{P}_{\text{ext}}$ **do**
  6:    $z_m^{I,\text{ext}} \leftarrow E_I(x_m^{I,\text{ext}})$, $z_m^{T,\text{ext}} \leftarrow E_T(x_m^{T,\text{ext}})$                           ▷ Eq. 2
  7: **end for**
  8: **// Cross-modal bidirectional mapping**
  9: Train $f_{I \to T}, f_{T \to I}$ on internal pairs $(z_i^{I,\text{int}}, z_i^{T,\text{int}})$                        ▷ Eq. 3
10: **// Clustering-based knowledge filtering (image modality)**
11: Run $K$-means on $\{z^{I,\text{int}}\}$ and $\{z^{I,\text{ext}}\}$ to obtain centroids $\{\mu_p^{I,\text{int}}\}_{p=1}^K$, $\{\mu_q^{I,\text{ext}}\}_{q=1}^K$.    ▷ Eq. 4
12: **for** each external cluster $q = 1, \dots, K$ **do**
13:    $p_q^I \leftarrow \arg\max_p \cos(\mu_q^{I,\text{ext}}, \mu_p^{I,\text{int}})$                                        ▷ Eq. 5
14:    $s_{\text{cent}}^I \leftarrow \cos(\mu_q^{I,\text{ext}}, \mu_{p_q^I}^{I,\text{int}})$                                          ▷ Eq. 6
15:    **if** $s_{\text{cent}}^I < \tau_{\text{cent}}$ **then**
16:        **drop** cluster $q$
17:    **else**
18:        keep cluster $q$, add to $\mathcal{C}_{\text{keep}}^I$
19:    **end if**
20: **end for**
21: **for** each cluster $c_q^{I,\text{ext}} \in \mathcal{C}_{\text{keep}}^I$ **do**
22:    **for** each sample $z_m^{I,\text{ext}} \in c_q^{I,\text{ext}}$ **do**
23:        $s_{\text{inst}}^I \leftarrow \cos(z_m^{I,\text{ext}}, \mu_{p_q^I}^{I,\text{int}})$                                ▷ Eq. 6
24:        **if** $s_{\text{inst}}^I < \tau_{\text{inst}}$ **then**
25:            **drop** $z_m^{I,\text{ext}}$
26:        **else**
27:            keep $z_m^{I,\text{ext}}$
28:        **end if**
29:    **end for**
30: **end for**
31: Repeat the same procedure for text embeddings $(z^{T,\text{int}}, z^{T,\text{ext}})$ to obtain $\mathcal{C}_{\text{keep}}^T$ and $\mathcal{Z}_{\text{keep}}^T$. (see Appendix A.2)
32: **// Cross-domain alignment (Procrustes, modality-specific)**
33: Build image MNN pairs $\{(z_i^{I,\text{int}}, z_m^{I,\text{ext}})\}$ and text MNN pairs $\{(z_i^{T,\text{int}}, z_m^{T,\text{ext}})\}$ using CSLS.
34: Solve $W_I^* = \arg\min_{W \in O(d)} \|Z_{\text{int}}^I - Z_{\text{ext}}^I W\|_F^2$, $W_T^* = \arg\min_{W \in O(d)} \|Z_{\text{int}}^T - Z_{\text{ext}}^T W\|_F^2$    ▷ Eq. 9
35: Iteratively refine $W_I^*, W_T^*$ until convergence $\|W^{(r)} - W^{(r-1)}\|_F < \epsilon$.
36: **// Knowledge base construction**
37: Project external embeddings: $\tilde{Z}_{\text{ext}}^I = Z_{\text{ext}}^I W_I^*$, $\tilde{Z}_{\text{ext}}^T = Z_{\text{ext}}^T W_T^*$                  ▷ Eq. 10
38: $\mathcal{K}_{\text{int}} = \{x^I, z^I, x^T, z^T\}$; $\mathcal{K}_{\text{ext}} = \{x^{I,\text{ext}}, \tilde{z}^{I,\text{ext}}, x^{T,\text{ext}}, \tilde{z}^{T,\text{ext}}\}$            ▷ Eq. 11
39: $\mathcal{K} = \mathcal{K}_{\text{int}} \cup \mathcal{K}_{\text{ext}}$                                                             ▷ Eq. 12
40: **return** $\mathcal{K}, f_{I \to T}, f_{T \to I}$

---

## A.3   ANALYSIS OF CLUSTERING-BASED KNOWLEDGE FILTERING

**Imbalanced clusters.** In small or imbalanced clusters, outliers may severely bias the centroid $\mu_c$, rendering centroid-based decisions unreliable. For a cluster $c$ with size $|c|$ and sample embeddings $\{z(x) \mid x \in c\}$, its centroid is $\mu_c$. We compute the average cluster radius as

$$u_c = \frac{1}{|c|} \sum_{x \in c} \|z(x) - \mu_c\|_2, \tag{27}$$

---

**Algorithm 2** The Workflow of Multi-Modal Fusion Retrieval and Candidate Selection

---

**Require:** Knowledge base $\mathcal{K}$; encoders $E_I, E_T$; maps $f_{I \to T}, f_{T \to I}$; generators $G_{I2T}$ (image→text), $G_{T2I}$ (text→image); retrieval size $k$; fusion weight $\alpha$; number of candidates $n$; weights $\lambda_1, \lambda_2$
**Ensure:** Completed internal dataset $\widehat{\mathcal{D}}$

1: // **Image → Text**
2: $z_j^I = E_I(x_j^I); \quad \hat{z}_j^T = f_{I \to T}(z_j^I)$                  ▷ Eq. 13
3: **for** candidate $r$ in knowledge base $\mathcal{K}$ **do**
4: $\quad s_{\text{img}}(r) \leftarrow \cos(z_j^I, z_r^I)$                  ▷ Eq. 14
5: **end for**
6: $\sigma \leftarrow \text{argsort}_r(-s_{\text{img}}(r), r); \quad \mathcal{A} \leftarrow \{\sigma(1), \ldots, \sigma(k)\}$                  ▷ Eq. 15–16
7: **for** $r \in \mathcal{A}$ **do**
8: $\quad \text{sim}_{\text{fuse}}(r) \leftarrow \alpha \cos(z_j^I, z_r^I) + (1 - \alpha) \cos(\hat{z}_j^T, z_r^T)$                  ▷ Eq. 17
9: **end for**
10: $\pi \leftarrow \text{argsort}_{r \in \mathcal{A}}(-\text{sim}_{\text{fuse}}(r), r)$                  ▷ Eq. 18
11: Reorder $\mathcal{A}$ and select top-1 caption $x_{r^\star}^T$
12: Build prompt $\mathcal{P}$
13: Generate $n$ caption candidates $\{\tilde{x}_1^T, \ldots, \tilde{x}_n^T\} \leftarrow G_{I2T}(x_j^I, \mathcal{P})$                  ▷ Eq. 19
14: // **Text → Image**
15: $z_h^T = E_T(x_h^T); \quad \hat{z}_h^I = f_{T \to I}(z_h^T)$                  ▷ Eq. 29
16: **for** candidate $r$ in knowledge base $\mathcal{K}$ **do**
17: $\quad s_{\text{text}}(r) \leftarrow \cos(z_h^T, z_r^T)$                  ▷ Eq. 30
18: **end for**
19: $\sigma \leftarrow \text{argsort}_r(-s_{\text{text}}(r), r); \quad \mathcal{A} \leftarrow \{\sigma(1), \ldots, \sigma(k)\}$                  ▷ Eq. 30
20: **for** $r \in \mathcal{A}$ **do**
21: $\quad \text{sim}_{\text{fuse}}(r) \leftarrow \alpha \cos(z_h^T, z_r^T) + (1 - \alpha) \cos(\hat{z}_h^I, z_r^I)$                  ▷ Eq. 31
22: **end for**
23: Reorder $\mathcal{A}$ and select top-1 image $x_{r^\star}^I$
24: Generate $n$ image candidates $\{\tilde{x}_1^I, \ldots, \tilde{x}_n^I\} \leftarrow G_{T2I}(x_h^T, x_{r^\star}^I)$                  ▷ Eq. 32
25: // **Candidate Selection**
26: // Text candidates
27: $s_T(\tilde{x}_j^T) = \lambda_1 \cos(E_T(\tilde{x}_j^T), \hat{z}^T) + \lambda_2 \text{BLEU}(\tilde{x}_j^T)$                  ▷ Eq. 20
28: // Image candidates
29: $s_I(\tilde{x}_j^I) = \lambda_1 \cos(E_I(\tilde{x}_j^I), \hat{z}^I) - \lambda_2 \text{NIQE}(\tilde{x}_j^I)$                  ▷ Eq. 21
30: Select the candidate with the highest score as the final completion
31: **return** $\widehat{\mathcal{D}}$

---

and let $\bar{u}$ denote the average radius across all clusters. A cluster is regarded as *imbalanced* if $u_c > \beta \cdot \bar{u}$, with $\beta = 2$ in our experiments. For such clusters, we adopt more robust representatives instead of plain centroids. Specifically, we use the $\nu_c$, defined as

$$\nu_c = \arg \min_{x \in c} \sum_{y \in c} \|z(x) - z(y)\|_2, \tag{28}$$

which is the most central sample in $c$ and less sensitive to outliers.

**Sensitivity to thresholds.**   We further analyze the impact of $\tau_{\text{cent}}$ and $\tau_{\text{inst}}$. If the thresholds are set too high, only very close samples are retained, reducing coverage and harming downstream performance. If the thresholds are set too low, many noisy or irrelevant external samples are included, introducing excessive noise and again degrading performance. Our experiments show that performance improves as the thresholds increase up to a moderate range, but deteriorates once they are either too strict or too relaxed. Based on this observation, we set both thresholds to 0.7, which provides a stable trade-off between precision and recall.

### A.4   VISUALIZATION OF KNOWLEDGE ALIGNMENT

To illustrate the necessity and effectiveness of our knowledge filtering and alignment, we visualize the embeddings of the internal dataset (MSCOCO) and an external dataset (CC3M) using t-SNE before and after alignment.

**Before alignment.** As shown in Fig. 2, the embeddings from the two datasets form distinct clusters, reflecting the domain gap. Although semantically related (both are natural image–text corpora), the representations are not directly compatible.

**After alignment.** As shown in Fig. 2, after applying our knowledge filtering and alignment, the external embeddings are rotated into the internal space, and semantically similar samples from both datasets become much closer, indicating successful mitigation of domain shift.

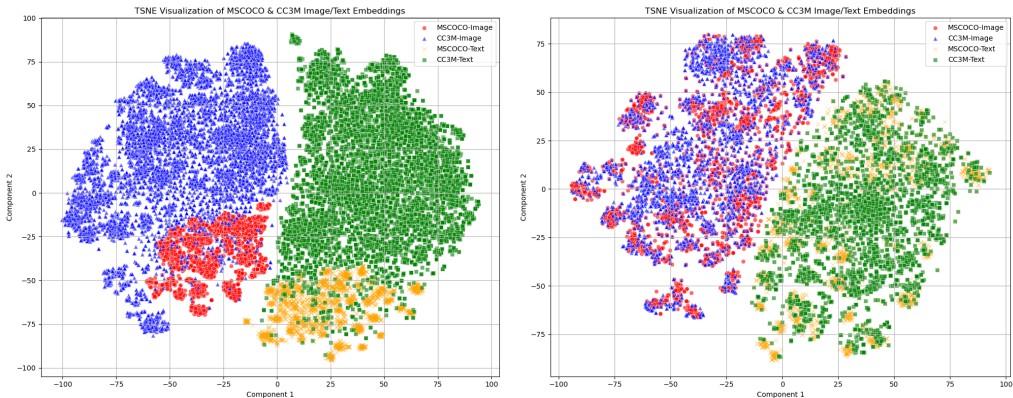

Figure 2: t-SNE visualization of MSCOCO and CC3M embeddings before (left) and after (right) alignment.

### A.5 MULTI-MODAL RETRIEVAL FOR TEXT-ONLY CASE

Given an unpaired text $x_h^T$, we extract its embedding $z_h^T = E_T(x_h^T)$ and obtain a pseudo-image embedding via the learned mapping:

$$\hat{z}_h^I = f_{T \to I}(z_h^T). \tag{29}$$

We perform intra-modal retrieval over $\mathcal{K}$ by text similarity and take the top-$k$ candidates:

$$s_{\text{text}}(r) = \cos(z_h^T, z_r^T), \quad \sigma = \underset{r}{\text{argsort}} \left( - s_{\text{text}}(r), r \right), \quad \mathcal{A} = \{\sigma(1), \dots, \sigma(k)\}, \, |\mathcal{A}| = k. \tag{30}$$

For $r \in \mathcal{A}$, we compute a fused score and pick the best exemplar image:

$$\text{sim}_{\text{fuse}}(r) = \alpha \, \cos(z_h^T, z_r^T) + (1 - \alpha) \cos(\hat{z}_h^I, z_r^I), \quad r^\star = \arg\max_{r \in \mathcal{A}} \text{sim}_{\text{fuse}}(r). \tag{31}$$

Finally, we feed the text $x_h^T$ and the retrieved reference image $x_{r^\star}^I$ into the text-to-image generator $G_{T2I}$ to produce the completed sample:

$$\tilde{x}^I = G_{T2I}(x_h^T, x_{r^\star}^I). \tag{32}$$

### A.6 DETAILS ON PROCRUSTES ALIGNMENT

In practice, all filtered external embeddings are considered during mutual nearest neighbor (MNN) matching. If an external embedding is linked to multiple internal embeddings, we retain only the pair with the highest CSLS score to ensure a one-to-one mapping. When the internal and external datasets differ greatly in size, we randomly subsample the larger side to balance the number of pairs used for alignment. Since Procrustes alignment is based on one-to-one MNN pairs, the number of effective seed pairs is bounded by the size of the internal dataset. Although this means that at most $|\mathcal{D}|$ external embeddings are directly used for learning the mapping, having a large external pool remains beneficial as it increases the chance of finding reliable MNN pairs. After learning the alignment from these one-to-one pairs, the resulting Procrustes transformation is applied to the entire external knowledge base, projecting all external embeddings into the internal space for subsequent retrieval.

## A.7 THEORETICAL DERIVATION OF ORTHOGONAL ALIGNMENT

The goal of cross-domain alignment is to map external embeddings $Z_{\text{ext}} \in \mathbb{R}^{n \times d}$ into the internal embedding space $Z_{\text{int}} \in \mathbb{R}^{n \times d}$ while preserving geometric structure. Formally, this corresponds to solving the orthogonal Procrustes problem defined in Eq. 9. For clarity, the following derivation is presented without modality superscripts, but applies equally to both image ($Z^I$) and text ($Z^T$) embeddings.

**Closed-form solution.** Expanding the Frobenius norm, the optimization is equivalent to

$$\max_{W \in O(d)} \text{Tr}(W^\top M), \quad \text{where } M = Z_{\text{ext}}^\top Z_{\text{int}}.$$

Let the singular value decomposition (SVD) of $M$ be

$$M = U\Sigma V^\top,$$

with $U, V \in \mathbb{R}^{d \times d}$ orthogonal and $\Sigma \in \mathbb{R}^{d \times d}$ diagonal. The optimal alignment matrix is

$$W^* = UV^\top,$$

which yields the aligned external embeddings as given in Eq. 10.

**Iterative refinement.** Since exact correspondence between $Z_{\text{ext}}$ and $Z_{\text{int}}$ is unknown, we refine $W^*$ iteratively following (Lample et al., 2018): (i) update mutual nearest neighbor (MNN) pairs using CSLS on the current aligned embeddings; (ii) recompute $W^*$ via the SVD step above; (iii) repeat until convergence, i.e.,

$$\|W^{(s)} - W^{(s-1)}\|_F < \epsilon,$$

for a small threshold $\epsilon$, where $s$ indexes the iteration.

**Domain-consistent embedding space.** After convergence, internal and external embeddings are merged into a unified space. The internal embeddings remain fixed as the anchor space (standardized by their own mean and scale), while the learned mapping $W^*$ is applied solely to external embeddings, ensuring domain-consistent representations.

## A.8 DATASET SPLITS AND IMPLEMENTATION DETAILS.

**Dateset Splits.** From MSCOCO, we sample 10k training instances and construct a modality-missing dataset where each modality has a 35% missing rate, resulting in a total missing rate of 70%. For Flickr30K, we use 5k training instances with a 20% missing rate per modality, yielding an overall missing rate of 40%. For RSICD, all 8,743 training samples are utilized with a 20% missing rate per modality, corresponding to a total missing rate of 40%. Since both CC3M and NWPU-Caption are large-scale, we randomly sample 10k entries from each to construct the external knowledge base.

**Implementation Details.** For knowledge alignment, we cluster embeddings with KMeans ($k = 128$), apply thresholds $\tau_{\text{cent}}, \tau_{\text{inst}} = 0.7$, and refine the alignment using Procrustes analysis (5 iterations). During knowledge-constrained completion, we retrieve top-$k = 10$ candidates and $\alpha = 0.5$ for conditioning. For candidate selection, we set $\lambda_1 = 0.7$ and $\lambda_2 = 0.3$ to balance semantic similarity and quality scores.

## A.9 COMPLEXITY ANALYSIS

We analyze the asymptotic cost of each component. Let $N$ be the number of entries (per modality) in the knowledge base, $d$ the embedding dimension, $K$ the number of clusters, $s$ the number of alignment refinements, $k$ the retrieval shortlist size, $n$ the number of generated candidates, and $P = |\mathcal{P}_{\text{int}}|$ the number of internal image–text pairs used to train the mapping.

**Cross-modal mapping** ($f_{I \to T}, f_{T \to I}$)**.** Training the lightweight MLPs on $P$ paired samples scales linearly in both $P$ and $d$. With $e$ epochs (and fixed-width MLP), the cost is $O(e\,P\,d^2)$.

**Knowledge filtering (clustering + thresholding).** $K$-means over $N$ embeddings per modality costs $O(NKd)$. Matching cluster centroids and computing centroid/instance similarities add at most $O(K^2d) + O(Nd)$, so the filtering step remains dominated by $O(NKd)$.

**Cross-domain alignment (MNN + Procrustes).** Let $N_f$ be the number of *filtered* embeddings that enter alignment (per modality). Per refinement step:

- Mutual nearest neighbor (MNN) construction with CSLS requires nearest-neighbor search across domains: $O(N_f^2 d)$.
- Procrustes: form $M = Z_{\text{ext}}^\top Z_{\text{int}}$ in $O(N_f d^2)$, then SVD on $d \times d$ in $O(d^3)$.

With $s$ refinements, the alignment cost is $O\big(s\,[\,N_f^2 d\ +\ N_f d^2 + d^3\,]\big)$ .

**Retrieval-based completion (two-stage retrieval + candidate scoring).** Given a query embedding (image or text): $O(Nd)\ +\ O(kd)\ +\ O(nd)$.

**Overall.** Summing the components yields $O(e\,P\,d^2)\ +\ O(NKd)\ +\ O\big(s\,[N_f^2 d + N_f d^2 + d^3]\big)\ +\ O(Nd + kd + nd)$ . In typical settings with fixed $d \ll N$ and small $k, n$, the dominant terms are $O(NKd)$ (filtering), the MNN search inside alignment, and the intra-modal retrieval term.

## A.10 EXPERIMENTS WITH STRONGER GENERATORS

To examine the effect of stronger generation models, we conducted additional experiments using enhanced image captioning and text-to-image models. As shown in Table 10, upgrading the generators (e.g., from BLIP2 + Stable Diffusion XL 1.0 to LLaVA-1.5 (Liu et al., 2023) + Juggernaut-XL) improves the performance of Direct Generation across both retrieval and captioning metrics. Notably, RAG4DMC achieves 49.1 / 49.4 R@1 and 117.9 CIDEr, outperforming the enhanced Direct Generation baseline by a clear margin. These results demonstrate that while stronger generators can enhance Direct Generation, RAG4DMC consistently provides additional gains on top of any generation backbone. This reinforces that our method improves modality completion beyond what generator quality alone can achieve.

Table 10: Performance of Direct Generation and RAG4DMC with different generation models.

| Method | Generation Models | I2T R@1 | T2I R@1 | I2T R@5 | T2I R@5 | CIDEr | BERTScore |
|---|---|---|---|---|---|---|---|
| Direct Generation | Blip2 and Stable Diffusion-XL | 41.4 | 43.9 | 75.4 | 76.3 | 112.2 | 92.2 |
| Direct Generation | LLaVA-1.5 and Juggernaut-XL | 45.8 | 47.2 | 78.3 | 78.0 | 113.0 | 92.2 |
| RAG4DMC | Blip2 and Stable Diffusion-XL | 46.6 | 47.5 | 79.0 | 79.7 | 117.2 | 92.5 |
| RAG4DMC | LLaVA-1.5 and Juggernaut-XL | **49.1** | **49.4** | **81.7** | **81.4** | **117.9** | **92.6** |

## A.11 ABLATION STUDY ON MULTI-MODAL FUSION SCORE

To evaluate whether the proposed multi-modal fusion score ($\text{sim}_{\text{fuse}}$) provides additional benefits beyond standard top-k retrieval, we conducted an ablation study comparing three settings: (i) Top-k: One exemplar is randomly selected from the retrieved top-k candidates and fed into the generator; (ii) Top-k-all: All $k$ retrieved exemplars are fed into the generator (i.e., no fusion and no selection); (iii) RAG4DMC: Our proposed fusion score is used to select the best exemplar among the $k$ candidates and feed it into the generator.

As shown in Table 11, Top-k-all performs the worst, likely because feeding all retrieved samples into the generator introduces excessive or noisy context. Top-k performs better, but RAG4DMC achieves the best results across all metrics, improving both retrieval (e.g., +0.6 R@1 over Top-k) and captioning (e.g., +1.0 CIDEr). These results demonstrate that the fused score effectively selects the most semantically aligned exemplar, yielding consistent performance gains over naive top-k retrieval strategies.

## A.12 ABLATION STUDY ON CROSS-DOMAIN ALIGNMENT ITERATIONS

To evaluate the effect of varying the number of cross-domain alignment iterations, we conducted experiments with 3, 4, and 5 iterations. As shown in Table 12, varying the number of iterations

Table 11: Performance of RAG4DMC with different retrieval methods.

| Method | I2T R@1 | T2I R@1 | I2T R@5 | T2I R@5 | CIDEr | BERTScore |
|---|---|---|---|---|---|---|
| Top-k | 46.0 | 46.7 | 78.4 | 78.8 | 116.0 | 92.3 |
| Top-k-all | 43.7 | 45.4 | 76.2 | 75.8 | 115.5 | 92.2 |
| RAG4DMC | **46.5** | **47.3** | **78.9** | **79.5** | **117.0** | **92.5** |

from 3 to 5 results in only marginal performance improvements (e.g., I2T R@1: $46.2 \rightarrow 46.6$; CIDEr: $116.3 \rightarrow 117.2$), demonstrating that the method is not highly sensitive to this parameter.

Table 12: Performance of RAG4DMC with different number of cross-domain alignment iterations.

| Number of Iterations | I2T R@1 | T2I R@1 | I2T R@5 | T2I R@5 | CIDEr | BERTScore |
|---|---|---|---|---|---|---|
| 3 | 46.2 | 46.7 | 78.6 | 78.8 | 116.3 | 92.4 |
| 4 | 46.4 | 46.9 | 78.8 | 79.2 | 116.8 | 92.4 |
| 5 | **46.6** | **47.5** | **79.0** | **79.7** | **117.2** | **92.5** |

## A.13 ABLATION STUDY ON CANDIDATE SELECTION WEIGHTS ($\lambda_1$ AND $\lambda_2$)

To examine the effect of candidate selection weights, we conducted experiments by varying $\lambda_1$ and $\lambda_2$. As shown in Table 13, varying these weights leads to moderate but controlled changes in performance. Importantly, RAG4DMC continues to outperform baseline methods across all tested settings.

Table 13: Performance of RAG4DMC with different $\lambda_1$ and $\lambda_2$ values.

| $\lambda_1$ | $\lambda_2$ | I2T R@1 | T2I R@1 | I2T R@5 | T2I R@5 | CIDEr | BERTScore |
|---|---|---|---|---|---|---|---|
| 0.7 | 0.3 | 47.7 | 48.5 | 80.1 | 80.3 | 121.5 | 92.6 |
| 0.5 | 0.5 | 47.2 | 48.4 | 80.1 | 80.4 | 121.3 | 92.6 |
| 0.3 | 0.7 | 46.4 | 47.5 | 79.0 | 79.7 | 120.9 | 92.5 |

## A.14 SEMANTIC AND FIDELITY EVALUATION OF GENERATED DATA

To provide a more comprehensive assessment of the generated data quality, we conducted direct semantic and fidelity evaluations. Specifically, we computed CLIP-Similarity between the generated text and the original modality images, and we evaluated image fidelity using FID, comparing generated images with 10k real MSCOCO images. As shown in Table 14, RAG4DMC achieves higher CLIP-Similarity and substantially lower FID (Heusel et al., 2017) compared to Direct Generation, indicating that our method produces text that is more semantically aligned with the original images and generates images with significantly better distributional fidelity. These results confirm that RAG4DMC improves not only downstream performance but also the intrinsic quality of the generated modalities.

Table 14: Semantic and fidelity evaluation of generated data.

| Method | CLIP-Similarity | FID |
|---|---|---|
| Real Data | 0.3040 | – |
| Direct Generation | 0.2915 | 30.70 |
| RAG4DMC | 0.2995 | 27.92 |

## A.15 EFFECT OF DOMAIN DISTANCE ON PERFORMANCE

To study the effect of domain distance, we compare two external knowledge base choices in Table 15: CC3M (a large, generic caption dataset) and Flickr30K (a smaller dataset more similar in style to our internal datasets). Although Flickr30K is closer in domain to the internal datasets, CC3M yields better performance, suggesting that semantic diversity and scale of the external KB are more important than strict domain matching. Furthermore, the gains achieved with CC3M show that RAG4DMC is robust to moderate domain mismatch and does not require a perfectly curated or tightly aligned external corpus.

Table 15: Performance of RAG4DMC with different external knowledge bases.

| External KB | I2T R@1 | T2I R@1 | I2T R@5 | T2I R@5 | CIDEr | BERTScore |
|---|---|---|---|---|---|---|
| CC3M | 46.6 | 47.5 | 79.0 | 79.7 | 117.2 | 92.5 |
| Flickr30K | 44.7 | 45.2 | 77.4 | 77.5 | 112.5 | 92.3 |

A.16 VISUAL COMPARISON OF RAG4DMC

In this appendix, we present a visual comparison of RAG4DMC's performance in filling missing modalities. Fig. 3 shows the comparison between real captions and those generated by RAG4DMC when the image modality is missing. On the left, we display real images from the MSCOCO dataset with their corresponding captions. The middle column shows captions generated through Direct Generation, which tend to be brief and less detailed. On the right, RAG4DMC-generated captions are more descriptive and contextually relevant, demonstrating its ability to fill in missing details.

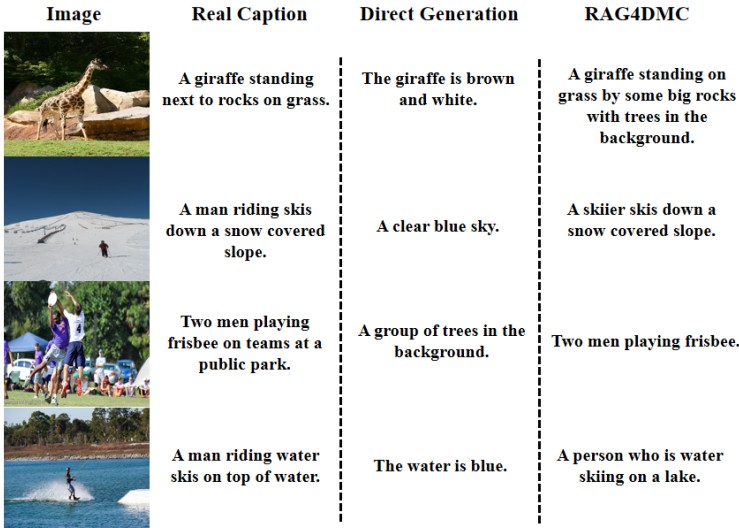

Figure 3: Comparison of generated captions when the image modality is missing.

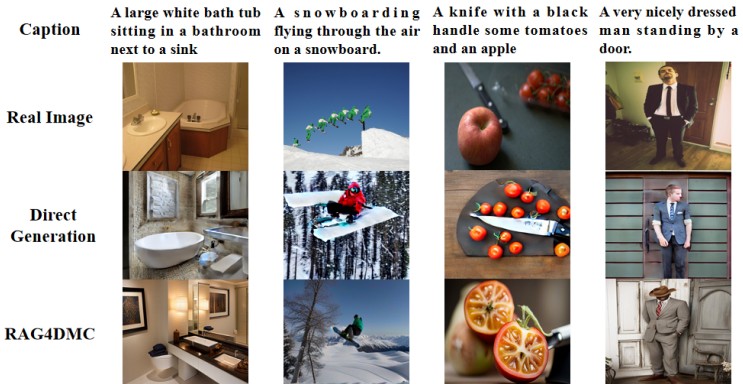

Figure 4: Comparison of generated images when the text modality is missing.

Fig. 4 illustrates RAG4DMC's performance in generating images when the text modality is missing. The left column shows real images with captions, while the middle column presents images generated through Direct Generation methods, often lacking detail or relevance. On the right, RAG4DMC generates more detailed and contextually accurate images, better reflecting the described scene.

These figures highlight the effectiveness of RAG4DMC in handling missing modalities and generating content that aligns with real scenes in both text and image modalities.

## A.17 USE OF LARGE LANGUAGE MODELS

During the preparation of this manuscript, we used OpenAI's GPT-5 as a language-assist tool to help improve the clarity and readability of certain paragraphs. All scientific ideas, experimental design, and data analysis were solely conceived and conducted by the authors. The outputs from the LLM were carefully reviewed and edited to ensure technical accuracy and consistency with our research findings.

