# OpenReview forum: "RAG4DMC: Retrieval-Augmented Generation for Data-Level Modality Completion"
_ICLR.cc/2026/Conference — ICLR 2026 Poster_

### Official Review · Reviewer_ZWWy · 2025-10-28

**Soundness:** 3
**Presentation:** 3
**Contribution:** 3
**Rating:** 6
**Confidence:** 3

**Summary:**

The paper tackles the Missing-Modality Completion (MMC) problem, where samples in a multi-modal dataset lack one or more modalities (e.g., an image without its paired caption). Instead of fine-tuning large generative models—often impractical due to limited complete samples, API-only access, or high cost—the authors propose RAG4DMC, a retrieval-augmented generation framework.

**Strengths:**

S1. Important practical problem: MMC occurs frequently in real-world pipelines, yet receives less attention than model-level fusion.
S2. Novel RAG formulation: combining internal and external KBs and explicitly tackling modality/domain shift via alignment + clustering is fresh in the MMC context.
S3. No fine-tuning of the large backbone: framework works even with “API-only” generative models, broadening applicability.
S4. Comprehensive evaluation: three datasets (general + domain-specific), multiple missing-rate settings, and downstream task benefits.
S5. Ablation on retrieval components and filtering validates each design choice.

**Weaknesses:**

W1. Scope limited to image-text dyads; unclear whether the approach scales to higher-order or fundamentally different modalities (e.g., audio, LiDAR).
W2. Generation quality is assessed only indirectly (via downstream tasks). Lack of direct fidelity/semantic metrics (e.g., CLIP-Similarity, FID) or human evaluation.
W3. External KB dependence: quality and domain bias of external data may dominate results; sensitivity analysis is brief.
W4. Computational footprint: retrieving from (possibly large) dual KB + generating multiple candidates can be expensive; runtime/memory costs are not reported.
W5. “First” claim could be overstated—earlier works have combined retrieval and generation for cross-modal reconstruction, though perhaps not under the MMC label; a clearer positioning is needed.
W6. Feature-alignment technique is only sketched; hyper-parameters and convergence behaviour are not well analysed.

**Questions:**

Q1. How does performance vary with the size and domain distance of the external KB? Please provide a scaling or ablation study.
Q2. What alignment method is used (e.g., Procrustes, MMD, contrastive loss)? Is it learned jointly with retrieval or pre-computed?
Q3. How many candidate completions are generated per query, and how sensitive are results to this number?
Q4. Could retrieval introduce harmful or biased content? Any safeguards or toxicity filtering applied?
Q5. What is the wall-clock cost (GPU hours) and latency per completion compared with direct generation?
Q6. Have you tested extremely high missing rates (e.g., 90 %) where internal KB is tiny? Does external data fully compensate?

---

> ### Author Response · Authors · 2025-11-21
>
> We thank the reviewer for the thoughtful and detailed feedback, which has helped us improve the clarity and completeness of the paper.
>
> **W1: Scope limited to image-text dyads; unclear whether the approach scales to higher-order or fundamentally different modalities (e.g., audio, LiDAR).**
>
> **Q:** We did not include other modalities experiments because current generative models for these modality pairs (e.g., video↔audio, skeleton↔depth, RGB↔optical flow) are not yet capable of producing reliable data-level completions, which are required by our RAG-based framework. Unlike images or text, generating even short, coherent video sequences or structured motion signals conditioned on another modality—together with retrieved exemplars—remains an open challenge. Existing video or audio generative models struggle with temporal consistency, fine-grained motion dynamics, and cross-modal alignment, making them unsuitable for controlled missing-modality reconstruction. Crucially, this limitation is not inherent to our method. RAG4DMC is modality-agnostic and requires only (i) a retrieval mechanism and (ii) a generator that can condition on the retrieved exemplar. As soon as reliable cross-modal generators for video, audio, skeleton, or depth become available, our framework can be directly applied to other modalities.
>
> **W2: Generation quality is assessed only indirectly (via downstream tasks). Lack of direct fidelity/semantic metrics (e.g., CLIP-Similarity, FID) or human evaluation.**
>
> **Q:** To provide a more comprehensive assessment of the generated data quality, we additionally conducted direct semantic and fidelity evaluations. Specifically, we computed CLIP-Similarity between generated text and the original modality images, and we evaluated image fidelity using FID, comparing generated images with 10k real MSCOCO images. As shown in Table 1, RAG4DMC achieves higher CLIP-Similarity and substantially lower FID compared with Direct Generation, indicating that our method produces text more semantically aligned with the original images and generates images with significantly better distributional fidelity. These results confirm that RAG4DMC improves not only downstream performance but also the intrinsic quality of the generated modalities. We have included this  evaluation in the revised manuscript.
>
> In addition to the quantitative comparisons, we have added multiple qualitative examples in the revised manuscript (see Figures 3 and Figures 4). These side-by-side visualizations compare the outputs of RAG4DMC with those of Direct Generation method for both missing-text and missing-image scenarios. The new qualitative results clearly illustrate that RAG4DMC produces more semantically coherent, fine-grained, and contextually aligned modality completions, especially in challenging cases where the baselines tend to generate overly generic or inconsistent content.
>
> Table 1. Semantic and fidelity evaluation of generated data.
> |Method|CLIP-Similarity|FID|
> |-|--|-|
> |Real Data|0.3040| -|
> |Direct Generation|0.2915|30.70|
> |RAG4DMC| 0.2995|27.92|
>
> **W3:  External KB dependence: quality and domain bias of external data may dominate results; sensitivity analysis is brief.**
>
> **Q:** We agree that external knowledge should ideally share some semantic relevance with the internal dataset. However, our external datasets are only loosely related to the internal domains. In our experiments, both MSCOCO and Flickr30K use CC3M as the external data source. The same external KB (CC3M) leads to performance improvements on both MSCOCO and Flickr30K. This indicates that: performance is not dominated by the external dataset, the proposed dual-KB structure and cross-domain alignment can mitigate domain bias, enabling effective transfer even when the domain match is not perfect.

---

> ### Author Response · Authors · 2025-11-21
>
> **W4: Computational footprint: retrieving from (possibly large) dual KB + generating multiple candidates can be expensive; runtime/memory costs are not reported.**
>
> **Q:**  In Appendix A.9, we provide a detailed theoretical complexity analysis for each component in our pipeline. As shown there, the dominant factors are the number of samples *N*, the number of clusters *k*, and the feature dimension *d*. We supplemented experiments measuring the runtime for each module under varying *N*, *k*, and *d*, on an NVIDIA GeForce RTX 3090 GPU.
>
> Our empirical results show that the cost is in fact moderate and well within a practical range. As shown in Table 1, under the largest tested configuration (*N* = 1k+10k, *k* = 128, *d* = 1024), the entire pipeline—including K-means, cross-domain alignment, and bidirectional MLP training—completes in ≈629 s (10.48 min), and retrieval remains extremely lightweight (0.0031 s). Thus, despite involving clustering and nearest-neighbor search, the end-to-end cost is clearly manageable in practice.
>
> Table 1 further breaks down where the computational cost arises.
> (i) K-means increases with *N* and *k*, consistent with its standard computational pattern;
> (ii) cross-domain alignment scales with *d* because nearest-neighbor search operates in the feature dimension;
> (iii) the bidirectional mapping (MLP) dominates the total cost but is largely unaffected by *N* and *k*, since it trains only on internal features and is primarily driven by *d*;
> and (iv) filtering and retrieval are negligible across all settings.
>
> This breakdown shows that all observed scaling behaviors are predictable and controlled, and none of the modules introduce unexpected or prohibitive overhead. Furthermore, users can adjust *N*, *k*, and *d* to balance efficiency and accuracy, enabling deployment under different computational budgets.
>
> Table 2 Runtime (seconds) of each module under different settings.
> |Setting|K-means|Filtering|Cross-domain Align|Cross-Modal Map|Retrieval |
> | --- | --: | ---: | ----: | ---: | ---: |
> |N=1k+5k, k=128, d=512|21.43|0.30|0.40 |405.17 |0.0012|
> |N=1k+10k, k=128, d=512|39.14|0.64|0.55| 407.14|0.0014|
> |N=1k+20k, k=128, d=512|112.87|1.35|0.99 |406.25|0.0017|
> |N=1k+10k, k=64, d=512|32.98|0.58|0.52|408.95 |0.0015|
> |N=1k+10k, k=256, d=512| 46.97|0.78|0.54 |408.98|0.0015|
> |N=1k+10k, k=128, d=768 |53.49|0.82|1.43|499.29 |0.0018|
> |N=1k+10k, k=128, d=1024|92.20|0.88|2.01|533.85|0.0031|
>
> **W5: "First" claim could be overstated—earlier works have combined retrieval and generation for cross-modal reconstruction, though perhaps not under the MMC label; a clearer positioning is needed.**
>
> **Q:** We agree that retrieval–generation ideas exist in prior multimodal work. However, these methods operate only at the latent/feature level—retrieved embeddings are injected into a model to approximate missing features for a specific task, and no raw modality is reconstructed. Our contribution differs in essence: RAG4DMC performs data-level multimodal completion, explicitly generating the raw missing modality (e.g., the actual image or text) rather than hidden representations. This enables task-agnostic, reusable multimodal data construction, which existing latent-level MMC or retrieval-augmented methods do not provide. We have refined the wording in the revised manuscript to clearly reflect this positioning as follows:
>
> "To our knowledge, RAG4DMC is the first RAG-based approach to achieve multimodal completion at the raw-data level, generating the actual missing modality instead of relying on latent-feature imputation. "

---

> ### Author Response · Authors · 2025-11-21
>
> **W6: Feature-alignment technique is only sketched; hyper-parameters and convergence behaviour are not well analysed.**
>
> **Q:**  To address the concern, we conducted additional analyses on the hyperparameters involved in the feature-alignment module, specifically examining the number of alignment iterations and the filtering threshold. The results are summarized in Tables 3 and 4. As shown in Table 3, varying the iteration count from 3 to 5 yields only small performance differences (e.g., I2T R@1: 46.2 → 46.6; CIDEr: 116.3 → 117.2). This indicates that the alignment module converges quickly in practice. Table 4 shows that performance remains stable when the filtering threshold varies between 0.6 and 0.8. Changes in retrieval and captioning metrics are minimal (e.g., CIDEr fluctuates within a narrow 117.0–117.2 range), indicating strong robustness to the threshold choice. Together, these analyses show that the feature-alignment process exhibits stable convergence and low hyperparameter sensitivity. We have incorporated these results into the revised paper for completeness.
>
> Table 3. Performance of RAG4DMC with different number of iterations.
> |Number of iterations|Image–Text Retrieval|Image Captioning| | | | |
> |-|-|-|-|-|-|-|
> | |I2T R@1|T2I R@1|I2T R@5|T2I R@5|CIDEr|BERTScore|
> |3|46.2|46.7|78.6|78.8|116.3|92.4|
> |4|46.4|46.9|78.8|79.2|116.8|92.4|
> |5|46.6|47.5|79.0|79.7|117.2|92.5|
>
> Table 4. Performance of RAG4DMC with different filtering thresholds.
> |Filtering Threshold|Image–Text Retrieval|Image Captioning| | | | |
> |-|-|-|-|-|--|-|
> | | I2T R@1|T2I R@1|I2T R@5|T2I R@5|CIDEr|BERTScore|
> |0.6|46.5|47.1| 79.0|79.0|117.0|92.4|
> |0.7|46.6|47.5| 79.0|79.7|117.2|92.5|
> |0.8|46.9|47.6|79.6 |79.9|117.2|92.5|
>
> **Q1: How does performance vary with the size and domain distance of the external KB? Please provide a scaling or ablation study.**
>
> **Q:** We conducted additional experiments to analyze how performance varies with (i) the size of the external knowledge base and (ii) its domain distance to the internal dataset.
>
> As shown in Table 5, we vary the number of external samples from 5k to 15k. The results show a monotonic improvement as the external KB grows, with diminishing returns as the size increases. This indicates that larger external KBs provide richer exemplars and better coverage, but our method does not rely on extremely large corpora to work effectively—even 5k external samples already yield competitive performance.
>
> To study the effect of domain distance, we compare two external KB choices in Table 6: CC3M (a large, generic caption dataset) and Flickr30K (smaller and more similar in style to our internal datasets). Although Flickr30K is closer in domain to the internal datasets, CC3M yields clearly better performance, suggesting that semantic diversity and scale of the external KB are more important than strict domain matching. At the same time, the gains we obtain with CC3M show that RAG4DMC is robust to moderate domain mismatch and does not require a perfectly curated or tightly aligned external corpus. We have included these scaling and domain-distance ablations in the revised appendix and clarified the discussion in the main text.
>
> Table 5. Performance of RAG4DMC with different number of the external knowledge base.
> |Number of external KB|I2T R@1|T2I R@1|I2T R@5|T2I R@5|CIDEr|BERTScore|
> |--|--|-|-|--|-|--|
> |5000|46.0|46.6|78.6|78.8|116.4|92.4|
> |10000|46.6|47.5|79.0|79.7|117.2|92.5|
> |15000|47.1|48.1|79.3| 80.1|117.5|92.7|
>
> Table 6. Performance of RAG4DMC with different external knowledge base.
> | External KB | I2T R@1 | T2I R@1 | I2T R@5 | T2I R@5 | CIDEr | BERTScore |
> |-|-|-|-|-|-|-|
> |CC3M |46.6|47.5|79.0|79.7|117.2| 92.5|
> |Flicker30|44.7|45.2|77.4| 77.5|112.5 |92.3 |
>
> **Q2: What alignment method is used (e.g., Procrustes, MMD, contrastive loss)? Is it learned jointly with retrieval or pre-computed?**
>
> **Q:** Our alignment module adopts a lightweight iterative nearest-neighbor–based alignment, which is classical Procrustes-style iterative refinement. Importantly, the alignment is not jointly learned with retrieval. Instead, we perform the cross-domain alignment offline, using the internal and external feature sets to compute the alignment transformation before building the knowledge bases. Once the aligned features are obtained, they are stored in the internal and external KBs, and no further optimization is performed during inference. Retrieval is then conducted directly on the aligned and pre-computed feature space. This design makes the alignment module efficient, stable, and fully decoupled from the retrieval and generation stages, and avoids additional training overhead. We have clarified this process more explicitly in the revised version of the paper.

---

> ### Author Response · Authors · 2025-11-21
>
> **Q3: How many candidate completions are generated per query, and how sensitive are results to this number?**
>
> **Q:** In the main paper, we evaluated the effect of generating different numbers of candidate completions for each query. As shown in Table 5, increasing the number of generated candidates n from 1 to 10 leads to smooth and moderate performance gains (e.g., I2T R@1: 46.6 → 47.7; CIDEr: 117.2 → 121.5) without causing instability. This demonstrates that RAG4DMC effectively identifies the best candidate among multiple generations, filtering out noisy outputsand ensuring that diversity translates into real performance gains.
>
> **Q4:  Could retrieval introduce harmful or biased content? Any safeguards or toxicity filtering applied?**
>
> **Q:**  In our framework, the retrieval component is safeguarded by a two-stage retrieval and reranking mechanism designed to suppress potentially harmful or biased content. Our fusion-based reranking score evaluates the retrieved candidates jointly across modalities, ensuring that only coherent, relevant, and high-quality items are selected as inputs to the generator. This reranking step effectively suppresses noisy or undesirable exemplars that could introduce bias into the final generations.
>
> **Q5: What is the wall-clock cost (GPU hours) and latency per completion compared with direct generation?**
>
> **Q:** To provide a clear comparison, we measured the end-to-end latency of generating a single missing modality using both Direct Generation and our RAG4DMC pipeline. All experiments were conducted using BLIP2 and Stable Diffusion-XL on an NVIDIA RTX 3090 GPU. Direct Generation requires 0.26 s for one text generation and 5.42 s for one image generation. In contrast, RAG4DMC takes 0.32 s for one text and 6.90 s for one image, including both retrieval and generation. This indicates that our retrieval-augmented procedure adds only a small overhead (approximately 0.06 s for text and 1.48 s for image), while yielding substantially more accurate and coherent completions.
>
> **Q6: Have you tested extremely high missing rates (e.g., 90 %) where internal KB is tiny? Does external data fully compensate?**
>
> **Q:** We conducted an additional experiment with a 90% missing rate, where the internal KB becomes very small. As shown in Table 7, RAG4DMC still produces reasonable completions (e.g., I2T R@1 = 37.2, CIDEr = 107.1) despite the drastic reduction of internal data. These results show that the external KB does provide meaningful compensation when internal data is severely limited. However, performance inevitably degrades compared with moderate missing rates, since some domain-specific information cannot be fully recovered from external data alone. This aligns with our motivation for dual KB construction: the external KB offers broad semantic coverage, while the internal KB contributes domain-specific grounding, and both are necessary for optimal performance.
>
> Table 7. Performance of RAG4DMC with different filtering thresholds.
> |Missing Rate|I2T R@1|T2I R@1|I2T R@5|T2I R@5|CIDEr|BERTScore|
> |-|-|-|-|-|-|-|
> |0.9|37.2|35.1|71.0|70.4|107.1|92.3|

---

> ### Author Response · Authors · 2025-11-27
>
> Dear Reviewer,
>
> As the author-reviewer discussion period will end soon, we would appreciate it if you could check our response to your review comments. This way, if you have further questions and comments, we can still reply before the author-reviewer discussion period ends. Thank you very much for your time!

---

### Official Review · Reviewer_MkeP · 2025-10-31

**Soundness:** 3
**Presentation:** 3
**Contribution:** 3
**Rating:** 6
**Confidence:** 4

**Summary:**

This paper proposes RAG4DMC to address the problem of missing modalities in multimodal datasets. Existing generation-based methods suffer from low reliability, while simple RAG-based approaches are ineffective due to noise and domain gaps in external data. To overcome these issues, the authors construct a dual knowledge base by combining internal complete data with external public data. They align and integrate the semantic spaces through cross-modal mapping, clustering-based filtering, and Procrustes alignment. For missing samples, a two-stage fusion retrieval is performed to find highly relevant examples, which are then used with BLIP2 and Stable Diffusion to generate restoration candidates. Finally, CLIP cosine similarity, BLEU, and NIQE metrics are used to select the most semantically consistent and high-quality result.

Experiments conducted on MSCOCO, Flickr30K, and RSICD demonstrate that RAG4DMC outperforms all baseline methods. Notably, as the missing rate increases, the performance gap between RAG4DMC and other methods widens, proving its robustness under data-deficient conditions. The authors further extend the concept of RAG beyond text generation to data-level multimodal restoration, showing that the generated complete datasets significantly enhance the performance of downstream models such as CLIP and LLaVA

**Strengths:**

- One of the key strengths of this paper is that it effectively overcomes the generalization limitations of fine-tuning approaches that rely on pre-trained generative models or a small number of fully observed samples. Notably, this work is the first to apply RAG to the Missing Modality Completion problem, leveraging retrieval-based semantic grounding to achieve more reliable and consistent restoration.

- The proposed RAG4DMC framework simultaneously utilizes both an internal knowledge base and an external knowledge base, addressing domain shift between the two through feature alignment and clustering-based filtering. These alignment and filtering procedures reduce noise and unify the semantic space, thereby improving retrieval quality and overall restoration accuracy.

- Through a multi-modal fusion retrieval strategy, the model fuses information across images and text, enabling more fine-grained search results. Additionally, the candidate selection stage ensures semantically coherent outputs, making the restoration process far more stable than simple generation-based reconstruction and enabling faithful modality recovery grounded in real data distribution.

- Even under various missing-ratio settings, downstream models trained on restored data from RAG4DMC achieved consistent and significant performance gains on tasks such as image–text retrieval and image captioning. These results demonstrate that the proposed approach remains robust and effective even in realistic scenarios with severe data incompleteness.

**Weaknesses:**

- RAG4DMC consists of multiple stages cross-modal mapping, knowledge filtering, cross-domain alignment, and retrieval-based completion which collectively result in high computational complexity. When applied to large-scale datasets, both training and inference may become slow, indicating potential inefficiency in large-scale or real-time environments.

- The framework focuses only on image and text modalities, leaving its applicability to other modalities such as speech or depth insufficiently explored. Further experiments are needed to verify whether the proposed approach can be effectively extended to additional modalities.

- Existing MMC methods generally suffer from limited transferability in out-of-domain scenarios, often requiring retraining to maintain performance on new domains, which incurs high computational cost. Although RAG4DMC mitigates this issue through a dual knowledge base and cross-domain alignment, the bidirectional mapping is learned primarily from internal data, making it difficult to guarantee complete domain transfer. Moreover, due to its reliance on BLIP2 and Stable Diffusion, the generation quality may degrade in domains that differ significantly from the pre-training data, potentially resulting in unnatural or inaccurate outputs.

**Questions:**

- The generator used in the proposed framework appears to be implemented based on Stable Diffusion, but it is unclear which specific version was used.

---

> ### Author Response · Authors · 2025-11-21
>
> We thank the reviewer for the thoughtful and detailed feedback, which has helped us improve the clarity and completeness of the paper.
>
> **W1: RAG4DMC consists of multiple stages cross-modal mapping, knowledge filtering, cross-domain alignment, and retrieval-based completion which collectively result in high computational complexity. When applied to large-scale datasets, both training and inference may become slow, indicating potential inefficiency in large-scale or real-time environments.**
>
> **Q:** In Appendix A.9, we provide a detailed theoretical complexity analysis for each component in our pipeline. As shown there, the dominant factors are the number of samples *N*, the number of clusters *k*, and the feature dimension *d*. We supplemented experiments measuring the runtime for each module under varying *N*, *k*, and *d*, on an NVIDIA GeForce RTX 3090 GPU.
>
> Our empirical results show that the cost is in fact moderate and well within a practical range. As shown in Table 1, under the largest tested configuration (*N* = 1k+10k, *k* = 128, *d* = 1024), the entire pipeline—including K-means, cross-domain alignment, and bidirectional MLP training—completes in ≈629 s (10.48 min), and retrieval remains extremely lightweight (0.0031 s). Thus, despite involving clustering and nearest-neighbor search, the end-to-end cost is clearly manageable in practice.
>
> Table 1 further breaks down where the computational cost arises.
> (i) K-means increases with *N* and *k*, consistent with its standard computational pattern;
> (ii) cross-domain alignment scales with *d* because nearest-neighbor search operates in the feature dimension;
> (iii) the bidirectional mapping (MLP) dominates the total cost but is largely unaffected by *N* and *k*, since it trains only on internal features and is primarily driven by *d*;
> and (iv) filtering and retrieval are negligible across all settings.
>
> This breakdown shows that all observed scaling behaviors are predictable and controlled, and none of the modules introduce unexpected or prohibitive overhead. Furthermore, users can adjust *N*, *k*, and *d* to balance efficiency and accuracy, enabling deployment under different computational budgets.
>
> Table 1 Runtime (seconds) of each module under different settings.
> |Setting|K-means Clustering|Filtering|Cross-domain Alignment|Cross-Modal Mapping|Retrieval|
> |-|-: |-|-: |-: |-: |
> |N=1k+5k, k=128, d=512|21.43|0.30|0.40 |405.17|0.0012|
> |N=1k+10k, k=128, d=512|39.14|0.64|0.55| 407.14|0.0014|
> |N=1k+20k, k=128, d=512|112.87|1.35|0.99 |406.25|0.0017|
> |N=1k+10k, k=64, d=512|32.98|0.58|0.52|408.95|0.0015|
> |N=1k+10k, k=256, d=512| 46.97|0.78|0.54 |408.98|0.0015|
> |N=1k+10k, k=128, d=768 |53.49|0.82|1.43|499.29 |0.0018|
> |N=1k+10k, k=128, d=1024|92.20|0.88|2.01|533.85|0.0031|
>
> **W2: The framework focuses only on image and text modalities, leaving its applicability to other modalities such as speech or depth insufficiently explored. Further experiments are needed to verify whether the proposed approach can be effectively extended to additional modalities.**
>
> **Q:** We did not include other modalities experiments because current generative models for these modality pairs (e.g., video↔audio, skeleton↔depth, RGB↔optical flow) are not yet capable of producing reliable data-level completions, which are required by our RAG-based framework. Unlike images or text, generating even short, coherent video sequences or structured motion signals conditioned on another modality—together with retrieved exemplars—remains an open challenge. Existing video or audio generative models struggle with temporal consistency, fine-grained motion dynamics, and cross-modal alignment, making them unsuitable for controlled missing-modality reconstruction. Crucially, this limitation is not inherent to our method. RAG4DMC is modality-agnostic and requires only (i) a retrieval mechanism and (ii) a generator that can condition on the retrieved exemplar. As soon as reliable cross-modal generators for video, audio, skeleton, or depth become available, our framework can be directly applied to other modalities.

---

> ### Author Response · Authors · 2025-11-21
>
> **W3: Existing MMC methods generally suffer from limited transferability in out-of-domain scenarios, often requiring retraining to maintain performance on new domains, which incurs high computational cost. Although RAG4DMC mitigates this issue through a dual knowledge base and cross-domain alignment, the bidirectional mapping is learned primarily from internal data, making it difficult to guarantee complete domain transfer. Moreover, due to its reliance on BLIP2 and Stable Diffusion, the generation quality may degrade in domains that differ significantly from the pre-training data, potentially resulting in unnatural or inaccurate outputs.**
>
> **Q:** We address the concerns from two perspectives:
>
> (1) Cross-domain transferability. We agree that existing MMC methods often exhibit limited transferability to out-of-domain scenarios and require retraining. Our method mitigates this issue through the dual knowledge base + cross-domain alignment design. As shown in our experiments across both generic and professional domains（Table 1-Table 3 in paper), KFA-RAG (which constructs the dual knowledge base using our knowledge filtering and alignment strategy) consistently outperforms the naïve Combined-RAG (which directly merges internal and external datasets) on all three evaluated datasets. This demonstrates that our knowledge filtering and alignment mechanisms can effectively reduce domain bias, improving cross-domain transferability.
>
> (2) Model-agnostic design. The primary purpose of our framework is to reduce the hallucination inherent in direct generation. While our current instantiation uses BLIP2 and Stable Diffusion XL 1.0 Refiner (SD-XL 1.0), RAG4DMC itself is model-agnostic: both the caption generator and image generator can be replaced with any stronger or more domain-adapted models. This flexibility ensures that generation quality can improve as better models become available. As shown in Table 2, RAG4DMC consistently outperforms Direct Generation under both model-pipeline settings (BLIP2 + SD-XL 1.0 and LLaVA-1.5 + Juggernaut-XL). This demonstrates that RAG4DMC reduces hallucination and improves semantic fidelity regardless of the underlying generation backbone.
>
> Table 2. Performance of Direct Generation and RAG4DMC with different generation models.
> | Method|Generation Models |I2T R@1|T2I R@1|I2T R@5|T2I R@5|CIDEr|BERTScore|
> |-|-|-|-|-|--|-|--|
> |Direct Generation|Blip2 and SD-XL 1.0|41.4|43.9|75.4 |76.3|112.2|92.2|
> |Direct Generation|LLaVA-1.5 and Juggernaut-XL|45.8|47.2|78.3|78.0|113.0 |92.2|
> |RAG4DMC|Blip2 and SD-XL 1.0|46.6| 47.5| 79.0| 79.7 |117.2| 92.5|
> |RAG4DMC|LLaVA-1.5 and Juggernaut-XL|49.1| 49.4 |81.7| 81.4|117.9| 92.6 |
>
> **Q1：The generator used in the proposed framework appears to be implemented based on Stable Diffusion, but it is unclear which specific version was used.**
>
> **Q:** In our implementation, the image generator is based on Stable Diffusion XL 1.0 Refiner. We have explicitly stated the exact version in the revised manuscript as follows:
>
> "We use BLIP2 as the text generator and Stable Diffusion XL 1.0 Refiner as the image generator across all datasets."

---

> ### Author Response · Authors · 2025-11-27
>
> Dear Reviewer,
>
> As the author-reviewer discussion period will end soon, we would appreciate it if you could check our response to your review comments. This way, if you have further questions and comments, we can still reply before the author-reviewer discussion period ends. Thank you very much for your time!

---

### Official Review · Reviewer_i2QC · 2025-10-31

**Soundness:** 2
**Presentation:** 3
**Contribution:** 3
**Rating:** 6
**Confidence:** 2

**Summary:**

This paper introduces RAG4DMC, a novel framework that applies retrieval-augmented generation (RAG) to the task of data-level Missing Modality Completion (MMC). The core contribution is a sophisticated system that constructs a dual knowledge base from both in-domain and external datasets, employing techniques like feature alignment and clustering-based filtering to mitigate domain and modality shifts. The proposed multi-modal fusion retrieval, which combines intra-modal retrieval with cross-modal re-ranking, is a clever approach to guide the generation process more effectively. The experimental results on both general and domain-specific datasets demonstrate that this method not only produces more accurate and semantically coherent completions but also leads to significant improvements in downstream tasks like image-text retrieval and captioning.

**Strengths:**

1. The paper is among the first to systematically adapt the RAG paradigm for data-level Missing Modality Completion, moving beyond feature-level imputation to generate complete, usable data samples.

2. The dual-knowledge-base design, which leverages both internal and external data, is a key strength. The proposed methods for feature alignment and clustering-based filtering to handle domain and modality gaps are well-motivated and appear effective.

3. The two-stage multi-modal fusion retrieval process is a significant contribution. By first using precise intra-modal retrieval and then refining with cross-modal signals, the method effectively mitigates the inherent modality gap, leading to more semantically relevant context for the generator.

4. The authors have conducted extensive experiments on a variety of datasets (MSCOCO, Flickr30K, and the domain-specific RSICD), evaluating the impact on multiple downstream tasks. The inclusion of several well-designed baselines and ablation studies (e.g., KFA-RAG, Combined-RAG) effectively dissects the contribution of each component of their framework.

**Weaknesses:**

1. The overall framework has many components and hyperparameters (e.g., clustering parameters, thresholds, fusion weights). This complexity might make the system difficult to tune and reproduce. While the appendix provides some details, a more in-depth analysis of hyperparameter sensitivity would be beneficial.

2. The construction of the knowledge base, particularly the clustering, filtering, and nearest-neighbor search for alignment, could be computationally expensive for very large-scale internal and external datasets. The complexity analysis in the appendix is helpful, but the practical implications on massive datasets remain a potential concern.

3. While the quantitative results are strong, the paper could benefit from more qualitative examples. Showing more side-by-side comparisons of the generated modalities from RAG4DMC versus the baseline methods would provide more intuitive evidence of its superior performance in generating semantically coherent and high-fidelity data.

**Questions:**

N/A

---

> ### Author Response · Authors · 2025-11-21
>
> We thank the reviewer for the thoughtful and detailed feedback, which has helped us improve the clarity and completeness of the paper.
>
> **W1: The overall framework has many components and hyperparameters (e.g., clustering parameters, thresholds, fusion weights). This complexity might make the system difficult to tune and reproduce. While the appendix provides some details, a more in-depth analysis of hyperparameter sensitivity would be beneficial.**
>
> **Q:** We conducted additional analyses to evaluate hyperparameters' sensitivity and confirm the robustness of the overall design. Our framework primarily has the following parameters: (1) Cross-domain alignment iterations count: As shown in Table 1, varying the number of cross-domain alignment iterations from 3 to 5 yields only marginal performance differences (e.g., I2T R@1: 46.2 → 46.6; CIDEr: 116.3 → 117.2), demonstrating that the method is not highly sensitive to this parameter. (2) Filtering threshold: Table 2 shows that performance remains stable when the filtering threshold varies between 0.6 and 0.8. Changes in retrieval and captioning metrics are minimal (e.g., CIDEr fluctuates within a narrow 117.0–117.2 range), indicating strong robustness to the threshold choice. (3) Candidate selection weights: As shown in Table 3, varying λ _1 and λ _2 leads to moderate but controlled changes in performance, and importantly, RAG4DMC continues to outperform baseline methods under all tested settings. Together, these results demonstrate that RAG4DMC is robust across a broad range of key hyperparameters, and the framework does not require delicate tuning to achieve strong performance.
>
> Table 1. Performance of RAG4DMC with different number of iterations.
> |Number of iterations|Image–Text Retrieval|Image Captioning| | | | |
> |-|-|-|-|-|-|-|
> | |I2T R@1|T2I R@1|I2T R@5|T2I R@5|CIDEr|BERTScore|
> |3|46.2|46.7|78.6|78.8|116.3|92.4|
> |4|46.4|46.9|78.8|79.2|116.8|92.4|
> |5|46.6|47.5|79.0|79.7|117.2|92.5|
>
> Table 2. Performance of RAG4DMC with different filtering thresholds.
> |Filtering Threshold|Image–Text Retrieval|Image Captioning| | | | |
> |-|-|-|-|-|-|-|
> | |I2T R@1|T2I R@1|I2T R@5|T2I R@5|CIDEr|BERTScore|
> |0.6|46.5|47.1|79.0|79.0|117.0|92.4|
> |0.7|46.6|47.5|79.0|79.7|117.2|92.5|
> |0.8|46.9|47.6|79.6 |79.9|117.2|92.5|
>
> Table 3. Performance of RAG4DMC with different λ_1 and λ_2.
> |λ_1|λ _2|Image–Text Retrieval|Image Captioning| | | | |
> |-|-|-|-|-|-|-|-|
> | | |I2T R@1|T2I R@1|I2T R@5|T2I R@5|CIDEr|BERTScore|
> |0.7|0.3 |47.7|48.5|80.1|80.3 |121.5|92.6|
> |0.5|0.5 |47.2|48.4|80.1|80.4|121.3|92.6|
> |0.3|0.7|46.4|47.5|79.0|79.7|120.9|92.5|

---

> ### Author Response · Authors · 2025-11-21
>
> **W2: The construction of the knowledge base, particularly the clustering, filtering, and nearest-neighbor search for alignment, could be computationally expensive for very large-scale internal and external datasets. The complexity analysis in the appendix is helpful, but the practical implications on massive datasets remain a potential concern.**
>
> **Q:** In Appendix A.9, we provide a detailed theoretical complexity analysis for each component in our pipeline. As shown there, the dominant factors are the number of samples *N*, the number of clusters *k*, and the feature dimension *d*. We supplemented experiments measuring the runtime for each module under varying *N*, *k*, and *d*, on an NVIDIA GeForce RTX 3090 GPU.
>
> Our empirical results show that the cost is in fact moderate and well within a practical range. As shown in Table 1, under the largest tested configuration (*N* = 1k+10k, *k* = 128, *d* = 1024), the entire pipeline—including K-means, cross-domain alignment, and bidirectional MLP training—completes in ≈629 s (10.48 min), and retrieval remains extremely lightweight (0.0031 s). Thus, despite involving clustering and nearest-neighbor search, the end-to-end cost is clearly manageable in practice.
>
> Table 1 further breaks down where the computational cost arises.
> (i) K-means increases with *N* and *k*, consistent with its standard computational pattern;
> (ii) cross-domain alignment scales with *d* because nearest-neighbor search operates in the feature dimension;
> (iii) the bidirectional mapping (MLP) dominates the total cost but is largely unaffected by *N* and *k*, since it trains only on internal features and is primarily driven by *d*;
> and (iv) filtering and retrieval are negligible across all settings.
>
> This breakdown shows that all observed scaling behaviors are predictable and controlled, and none of the modules introduce unexpected or prohibitive overhead. Furthermore, users can adjust *N*, *k*, and *d* to balance efficiency and accuracy, enabling deployment under different computational budgets.
>
> Table 4 Runtime (seconds) of each module under different settings.
> |Setting|K-means Clustering|Filtering|Cross-domain Alignment|Cross-Modal Mapping|Retrieval|
> |-|-:|-|-:|-:|-:|
> |N=1k+5k, k=128, d=512|21.43|0.30|0.40 |405.17|0.0012|
> |N=1k+10k, k=128, d=512|39.14|0.64|0.55| 407.14|0.0014|
> |N=1k+20k, k=128, d=512|112.87|1.35|0.99 |406.25|0.0017|
> |N=1k+10k, k=64, d=512|32.98|0.58|0.52|408.95|0.0015|
> |N=1k+10k, k=256, d=512| 46.97|0.78|0.54 |408.98|0.0015|
> |N=1k+10k, k=128, d=768 |53.49|0.82|1.43|499.29 |0.0018|
> |N=1k+10k, k=128, d=1024|92.20|0.88|2.01|533.85|0.0031|
>
> **W3: While the quantitative results are strong, the paper could benefit from more qualitative examples. Showing more side-by-side comparisons of the generated modalities from RAG4DMC versus the baseline methods would provide more intuitive evidence of its superior performance in generating semantically coherent and high-fidelity data.**
>
> **Q:** In addition to the quantitative comparisons, we have added multiple qualitative examples in the revised manuscript (see Fig.3 and Fig.4). These side-by-side visualizations compare the outputs of RAG4DMC with those of Direct Generation method for both missing-text and missing-image scenarios. The new qualitative results clearly illustrate that RAG4DMC produces more semantically coherent, fine-grained, and contextually aligned modality completions, especially in challenging cases where the baselines tend to generate overly generic or inconsistent content. We believe these examples provide intuitive evidence that complements the quantitative results.

---

> ### Author Response · Authors · 2025-11-27
>
> Dear Reviewer,
>
> As the author-reviewer discussion period will end soon, we would appreciate it if you could check our response to your review comments. This way, if you have further questions and comments, we can still reply before the author-reviewer discussion period ends. Thank you very much for your time!

---

### Official Review · Reviewer_nBpJ · 2025-11-02

**Soundness:** 2
**Presentation:** 3
**Contribution:** 2
**Rating:** 4
**Confidence:** 4

**Summary:**

This paper introduces RAG4DMC, a retrieval-augmented generation framework for data-level Missing Modality Completion (MMC). RAG4DMC overcomes challenges of fine-tuning large models and limited domain-specific data by structuring knowledge for more effective completion, leading to significant gains in downstream tasks. RAG4DMC builds a dual knowledge base —one from comprehensive in-dataset samples and another from external public datasets. It enhances performance by using feature alignment and clustering-based filtering to better manage differences in modality and domain.

**Strengths:**

* First framework for RAG tailored specifically for data-level missing modality completion problem.
* Combination of internal complete samples with external public datasets to generate the dual knowledge base.
* Two-stage multi-modal fusion retrieval strategy that leverages both intra-modal precision and cross-modal cues via pseudo-embeddings, ensuring the retrieval provides semantically consistent and highly relevant context for generation.
* Outperforms all baselines across general domain (MSCOCO, Flickr30K) and domain-specific (RSICD) datasets highlighting effectiveness in diverse tasks under high missing rates.

**Weaknesses:**

* High computational complexity of the proposed knowledge base construction especially the K-means clustering algorithm and iterative nearest neighbor search during cross-domain alignment.
* In the setting it should be mentioned that the incomplete samples are encountered during the training phase (through missing rates) and the evaluation is performed on the complete samples.
* High sensitivity of the performance to specific thresholds associated with filtering and retrieval size.
* How does the quality of results associated with Direct generation baseline change with improvements in image caption generation models (for missing text) and text to image models (for missing image).
* The negative effects of missing modalities are increasingly significant in multimodal classification tasks such as audio-visual action recognition (for example, Mit-51, UCF-101, and Activity Net as examined in GTI-MM). This study does not address experiments related to audio-visual action recognition with varying rates of missing data.

**Questions:**

* Regarding Equations (20) and (21), were the fixed weights for candidate selection empirically optimized for both BLEU (image-to-text) and NIQE (text-to-image)? Additionally, is it appropriate to apply the same weighting scheme given the differences between these metrics?
* Under multi-modal fusion retrieval, it is not clear in terms of ablation if sim_{fuse} provides additional benefits over top-k retrieval results.

---

> ### Author Response · Authors · 2025-11-21
>
> We thank the reviewer for the thoughtful feedback, which has helped us improve the clarity and completeness of the paper.
>
> **W1: High computational complexity of the proposed knowledge base construction especially the K-means clustering algorithm and iterative nearest neighbor search during cross-domain alignment.**
>
> **Q:** In Appendix A.9, we provide a detailed theoretical complexity analysis for each component in our pipeline. As shown there, the dominant factors are the number of samples *N*, the number of clusters *k*, and the feature dimension *d*. We supplemented experiments measuring the runtime for each module under varying *N*, *k*, and *d*, on an NVIDIA GeForce RTX 3090 GPU.
>
> Our empirical results show that the cost is in fact moderate and well within a practical range. As shown in Table 1, under the largest tested configuration (*N* = 1k+10k, *k* = 128, *d* = 1024), the entire pipeline—including K-means, cross-domain alignment, and bidirectional MLP training—completes in ≈629 s (10.48 min), and retrieval remains extremely lightweight (0.0031 s). Thus, despite involving clustering and nearest-neighbor search, the end-to-end cost is clearly manageable in practice.
>
> Table 1 further breaks down where the computational cost arises.
> (i) K-means increases with *N* and *k*, consistent with its standard computational pattern;
> (ii) cross-domain alignment scales with *d* because nearest-neighbor search operates in the feature dimension;
> (iii) the bidirectional mapping (MLP) dominates the total cost but is largely unaffected by *N* and *k*, since it trains only on internal features and is primarily driven by *d*;
> and (iv) filtering and retrieval are negligible across all settings.
>
> This breakdown shows that all observed scaling behaviors are predictable and controlled, and none of the modules introduce unexpected or prohibitive overhead. Furthermore, users can adjust *N*, *k*, and *d* to balance efficiency and accuracy, enabling deployment under different computational budgets.
>
> Table 1 Runtime (seconds) of each module under different settings.
> |Setting|K-means Clustering|Filtering|Cross-domain Alignment|Cross-Modal Mapping|Retrieval|
> |-|-:|-|-:|-:|-:|
> |N=1k+5k, k=128, d=512|21.43|0.30|0.40 |405.17|0.0012|
> |N=1k+10k, k=128, d=512|39.14|0.64|0.55| 407.14|0.0014|
> |N=1k+20k, k=128, d=512|112.87|1.35|0.99 |406.25|0.0017|
> |N=1k+10k, k=64, d=512|32.98|0.58|0.52|408.95|0.0015|
> |N=1k+10k, k=256, d=512| 46.97|0.78|0.54 |408.98|0.0015|
> |N=1k+10k, k=128, d=768 |53.49|0.82|1.43|499.29 |0.0018|
> |N=1k+10k, k=128, d=1024|92.20|0.88|2.01|533.85|0.0031|
>
>
> **W2: In the setting it should be mentioned that the incomplete samples are encountered during the training phase (through missing rates) and the evaluation is performed on the complete samples.**
>
> **Q:** In the original submission, we provided the missing-rate setting for the training data in Appendix A.1, but did not state explicitly that the evaluation is conducted on complete samples. We have revised the Experimental Setup section in the main text to make this clear:
>
>  “In our experiments, incomplete samples are introduced in the training phase by applying predefined missing rates to each modality. The evaluation phase is conducted exclusively on complete samples.”

---

> ### Author Response · Authors · 2025-11-21
>
> **W3: High sensitivity of the performance to specific thresholds associated with filtering and retrieval size.**
>
> **Q:** The thresholds referenced by the reviewer include the cluster-level and instance-level filtering thresholds *τ*_cent and *τ*_inst, as well as the retrieval size *k* used to determine the number of initially retrieved candidates. To assess their impact, we conducted additional sensitivity experiments by varying *τ*_cent and *τ*_inst within a reasonable range (0.6–0.8). The results (Table 2) show that retrieval and captioning performance remains highly stable: the changes in R@1, R@5, CIDEr, and BERTScore are minimal, indicating that our filtering strategy is robust to these thresholds.
>
> For the retrieval size *k*, we provide a detailed sensitivity study in Table 6 of the manuscript. Varying *k* within a common range (3–10) consistently improves performance, as a slightly larger candidate pool increases the chance of retrieving relevant exemplars. Performance only drops when *k* becomes excessively large (e.g., 15), where less relevant samples begin to dilute the useful context.
> Notably, *k* is a standard hyper-parameter in RAG-style systems rather than something specific to our method, and prior work [1–3] reports the same trend: moderate *k* improves quality, while overly large *k* introduces noise, further confirming that this behavior is inherent to retrieval-augmented pipelines rather than a limitation of our design.
>
> [1] Jin B, et al. Long-context LLMs meet RAG: Overcoming challenges for long inputs in RAG. arXiv preprint arXiv:2410.05983, 2024.
>
> [2] Liu Q, et al. Rethinking and improving autoformalization: Towards a faithful metric and a dependency retrieval-based approach. ICLR 2025.
>
> [3] Hwang J, et al. Retrieval-augmented generation with estimation of source reliability. EMNLP 2025: 34267–34291.
>
> Table 2. Performance of RAG4DMC with different filtering thresholds.
> |Filtering Threshold|Image–Text Retrieval|Image Captioning| | | | |
> |-|-|-|-|-|--|-|
> |  |I2T R@1|T2I R@1|I2T R@5|T2I R@5|CIDEr|BERTScore|
> |0.6|46.5|47.1| 79.0|79.0|117.0|92.4|
> |0.7|46.6|47.5| 79.0|79.7|117.2|92.5|
> |0.8|46.9|47.6|79.6 |79.9|117.2|92.5|
>
> **W4: How does the quality of results associated with Direct generation baseline change with improvements in image caption generation models (for missing text) and text to image models (for missing image).**
>
> **Q:** Indeed, replacing the current captioning and text-to-image components in Direct Generation with stronger generators would likely improve its performance. This effect is natural given that the baseline depends solely on the prior knowledge encoded in the generator. We clarify that RAG4DMC is model-agnostic: both its captioning and image-generation modules can be freely replaced with newer or stronger models. Thus, both Direct Generation and RAG4DMC would improve. To examine this effect, we conducted additional experiments using stronger image captioning and text-to-image models. As shown in Table 3, upgrading the generators (e.g., from BLIP2 + SD-XL 1.0 to LLaVA-1.5 + Juggernaut-XL) indeed improves the performance of Direct Generation across both retrieval and captioning metrics. Importantly, RAG4DMC achieves 49.1 / 49.4 R@1 and 117.9 CIDEr, outperforming the improved Direct Generation baseline by a clear margin. These findings demonstrate that while stronger generators can enhance Direct Generation, RAG4DMC continually provides additional gains on top of any generation backbone, reinforcing that our method improves modality completion beyond what generator quality alone can achieve.
>
> Table 3. Performance of Direct Generation and RAG4DMC with different generation models.
> | Method|Generation Models |I2T R@1|T2I R@1|I2T R@5|T2I R@5|CIDEr|BERTScore|
> |-|-|-|-|-|--|-|--|
> |Direct Generation|Blip2 and SD-XL 1.0 |41.4|43.9|75.4 |76.3|112.2|92.2|
> |Direct Generation|LLaVA-1.5 and Juggernaut-XL|45.8|47.2|78.3|78.0|113.0 |92.2|
> |RAG4DMC|Blip2 and SD-XL 1.0 |46.6| 47.5| 79.0| 79.7 |117.2| 92.5|
> |RAG4DMC|LLaVA-1.5 and Juggernaut-XL|49.1| 49.4 |81.7| 81.4|117.9| 92.6 |

---

> ### Author Response · Authors · 2025-11-21
>
> **W5: The negative effects of missing modalities are increasingly significant in multimodal classification tasks such as audio-visual action recognition (for example, Mit-51, UCF-101, and Activity Net as examined in GTI-MM). This study does not address experiments related to audio-visual action recognition with varying rates of missing data.**
>
> **Q:** We did not include audio–visual action recognition experiments because current generative models for these modality pairs (e.g., video↔audio, skeleton↔depth, RGB↔optical flow) are not yet capable of producing reliable data-level completions, which are required by our RAG-based framework. Unlike images or text, generating even short, coherent video sequences or structured motion signals conditioned on another modality—together with retrieved exemplars—remains an open challenge. Existing video or audio generative models struggle with temporal consistency, fine-grained motion dynamics, and cross-modal alignment, making them unsuitable for controlled missing-modality reconstruction. Crucially, this limitation is not inherent to our method. RAG4DMC is modality-agnostic and requires only (i) a retrieval mechanism and (ii) a generator that can condition on the retrieved exemplar. As soon as reliable cross-modal generators for video, audio, skeleton, or depth beacome available, our framework can be directly applied to audio–visual action recognition with varying missing-data ratios.
>
> **Q1: Regarding Equations (20) and (21), were the fixed weights for candidate selection empirically optimized for both BLEU (image-to-text) and NIQE (text-to-image)? Additionally, is it appropriate to apply the same weighting scheme given the differences between these metrics?**
>
> **Q:** As noted in Appendix A.8, the weights in Eq. (20) and (21) were set to λ_1 = 0.7 andλ_2 = 0.3. To assess the robustness of this choice, we conducted an ablation study on weight sensitivity (see Table 4). The results show that performance remains relatively stable across a reasonable range of λ_1 and λ_2 values, indicating that our method is not overly sensitive to this hyperparameter.
>
> The reviewer is correct that BLEU and NIQE measure different aspects of quality. Our framework is therefore designed to allow different weighting schemes for text candidates and image candidates, meaning that BLEU and NIQE can adopt different weights rather than sharing a universal one. This flexibility enables the weighting strategy to better accommodate the characteristics of each modality. We have clarified this design rationale in the revised manuscript and included the ablation table to further support our choice.
>
> Table 4. Performance of RAG4DMC with different λ_1 and λ_2.
> |λ_1| λ _2 | Image–Text Retrieval | Image Captioning | | | | |
> |-|-|-|-|-|-|-|-|
> | | |I2T R@1|T2I R@1|I2T R@5|T2I R@5|CIDEr|BERTScore|
> |0.7|0.3 |47.7|48.5| 80.1| 80.3 |121.5|92.6|
> |0.5|0.5 |47.2|48.4| 80.1| 80.4|121.3|92.6|
> |0.3|0.7|46.4|47.5| 79.0| 79.7|120.9 |92.5|
>
> **Q2: Under multi-modal fusion retrieval, it is not clear in terms of ablation if sim_{fuse} provides additional benefits over top-k retrieval results.**
>
> **Q:** To evaluate whether the proposed multi-modal fusion score sim_fuse provides additional benefits beyond standard top-k retrieval, we conducted an ablation comparing three settings: (i)Top-k, where one exemplar is randomly selected from the retrieved top-k candidates and fed into the generator; (ii)Top-k-all, where all *k* retrieved exemplars are fed into the generator (i.e., no fusion and no selection); (iii) RAG4DMC, which uses our proposed fusion score to select the best exemplar among *k* candidates and feed it into the generator. As shown in Table 5, Top-k-all performs the worst, likely because feeding all retrieved samples into the generator introduces excessive or noisy context. Top-k performs better, but RAG4DMC achieves the best results across all metrics, improving both retrieval (e.g., +0.6 R@1 over Top-k) and captioning (e.g., +1.0 CIDEr). These results demonstrate that the fused score effectively selects the most semantically aligned exemplar, yielding consistent performance gains over naive top-k retrieval strategies. We have included this ablation and clarification in the revised manuscript.
>
> Table 5. Performance of RAG4DMC with different retrieval methods.
> |Method|Image–Text Retrieval|Image Captioning| | | | |
> |-|-|-|-|-|-|-|
> |  |I2T R@1|T2I R@1| I2T R@5 |T2I R@5 | CIDEr | BERTScore |
> |Top-k|46.0|46.7|78.4| 78.8|116.0| 92.3|
> |Top-k-all|43.7|45.4|76.2|75.8| 115.5| 92.2|
> |RAG4DMC|46.5|47.3|78.9| 79.5| 117.0|92.5|

---

> ### Author Response · Authors · 2025-11-27
>
> Dear Reviewer,
>
> As the author-reviewer discussion period will end soon, we would appreciate it if you could check our response to your review comments. This way, if you have further questions and comments, we can still reply before the author-reviewer discussion period ends. Thank you very much for your time!

---

### Author Response · Authors · 2025-12-03

Dear AC and Reviewers, to give an overview of our response and updates we have made to the paper, we summarize our response below:
- Reviewer nBpJ，i2QC，MkeP and ZWWy raised conserns on the computational cost.  Appendix A.9 presented a full complexity analysis showing that all modules scale predictably with N, k, and d, with no hidden expensive operations. Our runtime experiments confirmed this: even under the largest tested configuration, the offline pipeline finished in 10.5 mins, and retrieval took only 0.0031 s; lighter settings required as little as 7.1 mins, already delivering strong performance. Since knowledge-base construction is performed once offline and N,k,d are fully tunable, the overall computational cost remains practical and easily controlled.
- Reviewer nBpJ, i2QC and ZWWy asked for deeper hyperparameter sensitivity analysis. We conducted sensitivity experiments showing that primarily parameters are all robust within broad ranges: (i) Cross-domain alignment iterations: Varying iterations 3→5 yields only marginal differences (e.g., Avg R@1: 46.4→47.0; CIDEr: 116.3→117.2). (ii) Filtering thresholds: Performance remains stable across 0.6–0.8 (CIDEr stays within 117.0–117.2). (iii) Candidate selection weights: Varying weights results in controlled changes (e.g., Avg R@1: 47.0→48.1; CIDEr: 120.9→121.5), and RAG4DMC consistently outperforms baselines across all settings.
- Reviewer  i2QC and ZWWy requested direct fidelity/semantic metrics for generation quality evaluation. We added CLIP-Similarity and FID evaluations, where RAG4DMC achieved higher CLIP-Similarity and substantially lower FID than Direct Generation, confirming better semantic coherence and distributional fidelity. We also added qualitative visualizations in the revised manuscript (Figs.3-4), which visually showed clearer, more fine-grained, and more contextually aligned completions.
- Reviewer nBpJ noted that stronger generation backbones would improve the Direct Generation baseline. Our added experiments showed that although stronger backbones did boost Direct Generation, RAG4DMC improved as well and continued to outperform it.
- Reviewer nBpJ questioned whether BLEU and NIQE should share a fixed weighting scheme. We evaluated the effect of varying the weights and observed stable performance across settings (e.g., Avg R@1: 47.0→48.1; CIDEr: 120.9→121.5). We also clarified in the paper that text and image candidates could adopt distinct weighting schemes to better reflect their respective metrics.
- Reviewer nBpJ asked whether the fusion score provided benefits beyond top-k selection. Our added ablation comparing top-k and our fusion method showed that our fusion method achieved the best performance, confirming its advantage in selecting the most semantically aligned exemplar.
- Reviewer ZWWy raised concerns about the “first” claim, noting that earlier work had combined retrieval for MMC. We refined our claim in revised paper to reflect the correct distinction: prior retrieval-augmented MMC methods operated only at the latent or feature level without generating raw missing modalities. In contrast, RAG4DMC performs data-level multimodal completion, producing actual data rather than imputing embeddings.
- Reviewer ZWWy asked for performance scaling with external KB size and domain distance. We added experiments varying the external KB size (5k–15k), which showed monotonic improvement and demonstrated that even 5k samples yielded competitive performance. We also compared CC3M (large, generic) with Flickr30K (smaller, closer domain) and found CC3M performed better, indicating that semantic diversity was more important than strict domain matching.
- Reviewer ZWWy requested GPU-hour or latency per completion compared with direct generation. We measured end-to-end latency on an NVIDIA RTX 3090 using BLIP2 and Stable Diffusion-XL. Direct Generation required 0.26 s for text and 5.42 s for image, while RAG4DMC required 0.32 s and 6.90 s, adding only 0.06 s and 1.48 s of overhead respectively—yet producing substantially more accurate and coherent outputs.
- Reviewer ZWWy asked for performance under extremely high missing rates. We conducted an additional experiment at a 90% missing rate, and RAG4DMC still produced reasonable completions (Avg R@1 36.2; CIDEr = 107.1).
- Reviewer nBpJ, MkeP, and ZWWy asked whether the method could extend to other modality pairs. Current cross-modal generators (e.g., video–audio, skeleton–depth) were unsuitable for the exemplar-conditioned generation required by RAG, which reflected limitations of existing generative models rather than our framework. RAG4DMC is modality-agnostic, and once reliable generators for additional modalities become available, it can be directly applied.

We believe that we have thoroughly addressed all reviewer concerns, and we sincerely thank the reviewers for their insightful and constructive feedback, which has significantly improved the clarity and technical rigor of our work.

---

### Meta-Review · Area_Chair_edEK · 2026-01-07

**Summary:**

This work explores a retrieval-augmented generation framework for missing modality completion (MMC). The proposed method first constructs a dual knowledge base from complete in-domain examples and external public datasets. Then, a two-stage multi-modal fusion retrieval process is applied to obtain the semantic consistent samples. Finally, the retrieved examples help generate candidates for completion and a selection strategy is developed to sample the appropriate candidates.

**Reviewer Concerns:**

The major concerns of reviewers are mainly from the complexity of the system, where multiple components are introduced for the task, e.g., cross-modal mapping, clustering-based filter, iterative nearest neighbor search, etc. All reviewers have the concern about the efficiency of the system and the sensitivity of the hyper-parameters in the proposed method. Besides, Reviewer nBpJ, MkeP and ZWWy find that the scope of the work is limited on vision-text tasks while additional modalities can be more helpful for real tasks. The rebuttal provides more ablation experiments for individual component in the proposed method. The running time of different components is also demonstrated to mitigate the concerns.

**Reviewer Scores:**

Most reviewers had positive scores before rebuttal. The additional experiments in rebuttal can help mitigate partial concerns from Reviewer nBpJ, who had a negative initial score. Although the common concern about limited modalities studied in this work remains unsolved after rebuttal, the current progress on vision-text tasks may still be valuable for future research.

---

### Decision · Program_Chairs · 2026-01-26

Accept (Poster)